# The Effect of Skeletal Muscle Oxygenation on Hemodynamics, Cerebral Oxygenation and Activation, and Exercise Performance during Incremental Exercise to Exhaustion in Male Cyclists

**DOI:** 10.3390/biology12070981

**Published:** 2023-07-10

**Authors:** Evgenia D. Cherouveim, Panagiotis G. Miliotis, Maria D. Koskolou, Konstantina Dipla, Ioannis S. Vrabas, Nickos D. Geladas

**Affiliations:** 1Division of Sports Medicine and Biology of Exercise, School of Physical Education and Sports Science, National and Kapodistrian University of Athens, 17237 Athens, Greece; 2Laboratory of Exercise Physiology and Biochemistry, School of Physical Education and Sports Science at Serres, Aristotle University of Thessaloniki, 62122 Serres, Greece

**Keywords:** incremental exercise test, muscle blood flow restriction, muscle oxygenation, cerebral oxygenation, cardiorespiratory fitness, cyclists, cerebral activity

## Abstract

**Simple Summary:**

This study aimed to elucidate whether exercise intolerance with muscle blood flow restriction is located within the central nervous system or the heart, namely whether cardiac response constitutes the crucial point precipitating exercise cessation. Muscle blood flow restriction (venous occlusion) during whole-body dynamic exercise compared to control condition, accelerates: (a) the rate of skeletal muscle deoxygenation, (b) the increase in systolic blood pressure (c) the rating of perceived exertion; and (d) cerebral activation, without these variables will be different at exhaustion point between the two experimental conditions, despite a marked decrease in maximal exercise time and maximal aerobic output. Maximal cardiac responses (i.e., heart rate, stroke volume, and cardiac output) were also significantly limited with muscle blood flow restriction. These findings suggest that skeletal muscle oxygenation levels in combination with augmented blood pressure response and cerebral activation may be important determinants in setting the limits of exercise performance. Specifically, when the aforementioned variables reach a predetermined maximal level, exercise ability might be limited despite the fact that cardiac performance never ends up at peak values. Muscle blood flow restriction might simulate pathological conditions such as heart failure, hypertension, and peripheral vascular diseases, which are all characterized by skeletal muscle perfusion impairments, and, thus provide insights into exercise intolerance in health and disease.

**Abstract:**

This study aimed to elucidate whether muscle blood flow restriction during maximal exercise is associated with alterations in hemodynamics, cerebral oxygenation, cerebral activation, and deterioration of exercise performance in male participants. Thirteen healthy males, cyclists (age 33 ± 2 yrs., body mass: 78.6 ± 2.5 kg, and body mass index: 25.57 ± 0.91 kg·m^−1^), performed a maximal incremental exercise test on a bicycle ergometer in two experimental conditions: (a) with muscle blood flow restriction through the application of thigh cuffs inflated at 120 mmHg (with cuffs, WC) and (b) without restriction (no cuffs, NC). Exercise performance significantly deteriorated with muscle blood flow restriction, as evidenced by the reductions in $\dot{V}$O_2max_ (−17 ± 2%, *p* < 0.001), peak power output (−28 ± 2%, *p* < 0.001), and time to exhaustion (−28 ± 2%, *p* < 0.001). Muscle oxygenated hemoglobin (Δ[O_2_Hb]) during exercise declined more in the NC condition (*p* < 0.01); however, at exhaustion, the magnitude of muscle oxygenation and muscle deoxygenation were similar between conditions (*p* > 0.05). At maximal effort, lower cerebral deoxygenated hemoglobin (Δ[HHb]) and cerebral total hemoglobin (Δ[THb]) were observed in WC (*p* < 0.001), accompanied by a lower cardiac output, heart rate, and stroke volume vs. the NC condition (*p* < 0.01), whereas systolic blood pressure, rating of perceived exertion, and cerebral activation (as assessed by electroencephalography (EEG) activity) were similar (*p* > 0.05) between conditions at task failure, despite marked differences in exercise duration, maximal aerobic power output, and $\dot{V}$O_2max_. In conclusion, in trained cyclists, muscle blood flow restriction during an incremental cycling exercise test significantly limited exercise performance. Exercise intolerance with muscle blood flow restriction was mainly associated with attenuated cardiac responses, despite cerebral activation reaching similar maximal levels as without muscle blood flow restriction.

## 1. Introduction

Maximal oxygen consumption ($\dot{V}$O_2max_) or maximal aerobic capacity is defined as the maximum volume of oxygen consumed per unit of time by cells at maximum effort [1]. The $\dot{V}$O_2max_ concept is widely used in sports and clinical settings. Specifically, it is referred to as an indicator of (a) cardiovascular and cardiopulmonary fitness [2], (b) aerobic capacity [3], (c) exercise training intensity quantification [4], (d) exercise training program effectiveness [5], (e) heart failure severity [6], and (f) morbidity and mortality in cardiovascular and metabolic diseases [7,8].

The limiting factors of $\dot{V}$O_2max_ have been consistently studied since 1924, but there is still controversy over which factor is the critical one. Exercise performance during incremental exercise to exhaustion relies on multiple and diverse facets which have been classified into two main categories: those of the oxygen delivery system, and those of the oxygen consumption system. The factors associated with the limitations of the oxygen delivery system include the (a) oxygen-carrying capacity of blood [9], (b) pulmonary diffusion capacity [10], (c) maximal cardiac output [11,12,13], (d) muscle blood flow [14,15], and (e) cerebral blood flow and oxygenation [16,17,18]; the limitations of the oxygen consumption system include the (a) muscle oxygen-diffusing capacity [19], (b) mitochondrial enzyme levels [20], and (c) muscle capillary density [21]. Furthermore, central command has been shown to contribute to the limitation of exercise capacity through modification of cerebral activation, and the rate of perceived exertion [22]. It is worth noting that group III and IV muscle afferents’ exaggerated feedback to the central nervous system has also been implied as a limiting factor [23,24,25]. 

Numerous studies have highlighted the contribution of the oxygen delivery system, specifically the ability of the cardiopulmonary system to transport O_2_ to the skeletal muscles, as the primary limiting factor of $\dot{V}$O_2max_. Indeed, the cardiovascular/anaerobic model of exercise performance states that the heart has a limited maximum pumping capacity (maximal cardiac output) for supplying blood; when this limit is attained, impaired oxygen delivery to active skeletal muscles leads to anaerobiosis, and as a consequence, the concentration of metabolic by-products inhibits muscle contraction, inducing fatigue [26,27,28]. Thus, cardiac function limitations lead to exhaustion and a decrease in muscle performance. However, the majority of such studies have been conducted during exercise in systemic hypoxia conditions, which modify arterial oxygen content, thus affecting arterial systemic oxygen availability and causing stress to the O_2_ delivery and consumption system. However, this approach impairs concomitantly cardiovascular, ventilatory, cerebrovascular, and peripheral hemodynamic responses’ causing an entanglement, the variability of which might limit exercise aerobic performance. 

In recent years, muscle blood flow restriction during resistance and aerobic exercise has been thoroughly applied in exercise training protocols as a novel intervention to enhance skeletal muscle and aerobic adaptations in healthy individuals, as well as in patients with chronic disease [29,30,31,32]. This intervention relies on the application of external pressure over the proximal limb musculature (upper and lower), with the intent to reduce arterial inflow, restrict venous blood flow, cause blood pooling in capacitance vessels, and accumulate metabolic by-products within the exercising skeletal muscles [33,34,35]. Thus, muscle blood flow restriction during exercise is thought to induce a reduction in skeletal muscle oxygenation (tissue hypoxemia), while unlike in hypoxia experiments, systemic oxygen availability is maintained. 

However, only one earlier study [36] conducted in our laboratory to has thus far investigated the effects of restricted skeletal muscle oxygenation, through the application of thigh cuffs, on maximal aerobic capacity; this study reported that exercise performance (as defined by $\dot{V}$O_2max_) and peak power output (PPO) were significantly reduced in incremental exercise. Furthermore, these results were accompanied by a similar level of skeletal muscle deoxygenation and perceptual exertion at task failure, despite the marked differences in exercise tolerance, whereas heart rate was significantly lower in the muscle blood flow restriction condition. Thus, the impairment in exercise tolerance with muscle blood flow restriction was only associated with a reduction in maximal heart rate, suggesting cardiac limitation to be the prime contributor to exercise intolerance during muscle blood flow restriction. However, it was not possible to untangle whether the lower maximal heart rate (HR_max_) was due to the lower PPO produced, or vice versa. This is a crucial question, because there is a long-standing debate about whether aerobic capacity is limited by the cardiovascular or central nervous system. Furthermore, in the aforementioned study [36], there were several limitations. Specifically, the sample size was small (six young male participants with an average exercise capacity of $\dot{V}$O_2max_: 42.9 ± 3.7 mL∙kg^−1^∙min^−1^), and the cardiovascular responses to incremental exercise to exhaustion were not thoroughly investigated except for the HR recording. In addition, the contribution of cerebral activation and oxygenation to exercise intolerance during incremental exercise with muscle blood flow restriction were not taken into consideration. A recent study by our laboratory (Cherouveim et al., 2021 [37]) examined the involvement of cerebral oxygenation and activation during muscle blood flow restriction at rest, and found that muscle oxygenation reduction through venous occlusion even at rest can be sensed by the central nervous system (CNS), probably via the feedback of group III/IV muscle afferents, thereby eliciting changes in cerebral oxygenation/activation. To our knowledge, the effect of muscle blood flow restriction on cerebral activation and oxygenation during whole-body dynamic exercise has not been investigated yet. 

Therefore, this study aimed to examine, in a novel and comprehensive manner, whether skeletal muscle oxygenation reduction via the application of thigh cuffs (venous occlusion) would affect more the beat-by-beat cardiac responses, the cerebral oxygenation (as assessed using near-infrared spectroscopy (NIRS)), or the cerebral activation (as assessed using EEG) during incremental exercise at exhaustion point in male cyclists. We hypothesized that compromising skeletal muscle oxygenation would augment cerebral activation earlier, reaching prematurely peak values, whereas maximal cardiac responses will never peak, and as a consequence, exercise tolerance would be limited during whole-body maximal exercise. 

## 2. Materials and Methods

### 2.1. Participants

The inclusion criteria were as follows: (1) only male participants; (2) recreational cyclists or triathletes with at least five years of specific cycling training; (3) nonsmokers; (4) no restraining cardiovascular, respiratory, musculoskeletal or neurological diseases; (5) age above 18 years; (6) no drug abuse or use of medications known to affect exercise performance; and (7) ability to follow the procedures of the current study and visit our laboratory a total of three times. All participants (*n* = 13) who met the inclusion criteria voluntarily participated in this study and completed the experimental procedures. Their mean (±SEM) age, body mass, stature height, body fat, and $\dot{V}$O_2max_ were 33 ± 2 yrs. (18–45 yrs.), 78.6 ± 2.5 kg (63.6–93.7 kg), 176.0 ± 1.9 cm (163–188 cm), 12.4 ± 1.5% (4.45–17.69%), and 50.5 ± 2.2 mL∙kg^−1^∙min^−1^ (39.24–65.17 mL∙kg^−1^∙min^−1^), respectively. Participants attended the laboratory after refraining from any heavy meals or caffeine-containing beverages for at least 2 h, and from any intense exercise for at least two days before testing. During the study, participants were instructed to maintain their normal diet and regular physical activity habits. Written informed consent was obtained from all participants after being informed of the experimental procedures, potential risks involved, and discomfort associated with muscle blood flow restriction. The experimental protocol was approved by the University’s Ethical Committee for human experimentation, and conformed to the Declaration of Helsinki, as revised in 2000.

### 2.2. Experimental Procedures

The participants visited the laboratory three times in total. During the first visit, the participants became familiar with the experimental procedures and underwent an evaluation of their pulmonary function. Next, participants joined the main experiment. The main experimental protocol consisted of an incremental exercise test to exhaustion on a cycle ergometer on two separate occasions (visits 2 and 3): (a) without (no cuffs, NC) and (b) with (with cuffs, WC) muscle blood flow restriction, separated by at least 2–3 days, in a randomized crossover design (Figure 1). The main experiments were carried out at the same time of day for each participant in a thermoneutral environment (temperature 21–22 °C and relative humidity 35–45%). 

### 2.3. Experimental Protocol 

After obtaining the anthropometric characteristics, the participant performed a standardized warm-up that consisted of 10 min cycling at 50% HR_max_ followed by stretching exercise, and then was equipped with the necessary non-invasive probes and sensors. Maximal voluntary isometric contraction (MVC) of the knee extensors was measured.

Baseline values were recorded for 5 min after stabilization of the hemodynamic parameters, while the participant was resting in a back-supporting chair, with their legs bent at the hip and knee (~90°). Thereafter, according to the experimental condition, thigh cuffs were inflated at 120 mmHg (or not) for 10 min, with the participant remaining in the same position. Next, the participant performed incremental exercise to exhaustion on an electronically braked cycle ergometer (Lode RH, Groningen, The Netherlands), either without (NC) or with (WC) muscle blood flow restriction. The initial workload was set at 30 W with increments of 30 W per min until volitional exhaustion (Figure 1). During exercise, participants were instructed to maintain pedaling rates at 80–90 rpm; verbal encouragement was provided throughout the last phases of each trial. $\dot{V}$O_2max_ was considered the highest mean value recorded during the last 15 s of exercise, as previously suggested [38], if at least three of the following criteria were satisfied: (1) identification of a plateau in oxygen uptake ($\dot{V}$O_2_) or an increase smaller than 150 mL·min^−1^, despite the increase in power output; (2) volitional exhaustion or/and inability to maintain a pedaling rate of 50 rpm, despite strong verbal encouragement; (3) a heart rate within 10% of the age-predicted maximum; and (4) a respiratory exchange ratio ≥1.10 based on the American Heart Association and American College of Sports Medicine guidelines [39]. PPO was defined as follows: PPO = PO_final_ + (t/60 × 30 Watts), where PO_final_ is the last completed workload and t is the number of seconds for which the final uncompleted workload was retained [40]. 

For muscle blood flow restriction, the thigh cuffs (18 cm width) were applied around the most proximal portion of each femur (thigh) of the participants, and immediately after recording the baseline values, within 2–3 s, the cuffs were manually inflated to 120 mmHg using mercury-in-rubber strain gausses, and were maintained until the end of the exercise protocol. The cuff occlusion pressure was controlled throughout the experimental protocol, and if needed, adjustments were made to keep it constant. Cuffs were connected to a previously calibrated analogical sphygmomanometer (Mac-Check, Anats, model 501, Japan), through which the requested levels of occlusion pressure could be easily adjusted and maintained on both thighs. The absolute occlusion pressure value of 120 mmHg is able to elicit approximately 55–65% blood flow restriction [41]. On the contrary, during the NC protocol, the participants performed the same experimental procedure without application of the thigh cuffs. 

### 2.4. Measurements 

#### 2.4.1. Muscle and Cerebral Oxygenation

A near-infrared spectrometer (NIRS; Oxymon Mk III, Artinis Medical Systems, Elst, The Netherlands) with two continuous wavelengths of near-infrared light (760 nm and 850 nm) was used to monitor changes in muscle and cerebral oxygenation continuously throughout experimental protocols. The theory, limitations, and reliability of measurements obtained through incremental exercise with this instrument have previously been mentioned [42]. Concentration changes in oxy- ([O_2_Hb]) and deoxy-hemoglobin ([HHb]) were calculated, providing information on the dynamic balance between O_2_ delivery and extraction in the underlying tissue [43,44]. Changes in total hemoglobin (Δ[THb]) were calculated as the sum of Δ[O_2_Hb] and Δ[HHb], which reflect regional microvascular blood volume changes [45,46], whereas changes in hemoglobin difference (Δ[HbDiff]) were calculated as the difference of Δ[HHb] and Δ[O_2_Hb] and used as an index of changes in tissue oxygenation [47]. 

For muscle oxygenation, the NIRS emitter and detector pair was attached to the middle portion of the left vastus lateralis muscle at the mid-thigh level and parallel to the long axis of the muscle (~15 cm above the proximal border of the patella, and ~5 cm lateral to the midline of the thigh). The cerebral NIRS probe was placed over the left prefrontal cortex, approximately 2 cm above the left eyebrow, and as laterally as possible to the longitudinal cerebral fissure. A flexible, plastic spacer with a fixed optode distance of 4.5 cm was used for the NIRS optodes to be held in place and secured to the skin. In addition, black elastic bandages were used to cover and stabilize the probe to eliminate movement artifacts and extraneous light interference. The placement of the NIRS probes was marked with indelible ink to facilitate subsequent attachments. Muscle and cerebral measurements were normalized to zero (*bias*) before recording baseline values to express the magnitude of muscle and cerebral deoxygenation at rest and maximal exercise, both with and without muscle blood flow restriction. 

#### 2.4.2. Cardiopulmonary Measurements 

Cardiovascular responses were continuously recorded noninvasively (Finometer 2003, FMS, Arnhem, The Netherlands), as previously described [48,49]. Systolic (SBP) and diastolic (DBP) blood pressure and heart rate (HR) were recorded beat-to-beat from the middle finger of the right hand using a photoplethysmometer (Finometer 2003, FMS, Arnhem, The Netherlands). During exercise, the participants placed and supported their arm at heart level and were instructed to keep their hands and fingers relaxed to obtain a strong pulse waveform from the right middle finger. Stroke volume (SV) was estimated through the Modelflow method [50]. Mean arterial pressure (MAP), cardiac output ($\dot{Q}$), and total peripheral resistance (TPR) were calculated using BeatScope software (version 1.a). The Finometer system was calibrated according to the manufacturer’s instructions. 

Gas exchange and ventilatory response were recorded continuously and breath by breath via open-circuit spirometry (Ultima CPX, MedGraphics, Saint Paul, MN, USA). The O_2_ and CO_2_ gas analyzers, as well as a pneumotachograph, were calibrated with two separate gas mixtures before each test: (a) 12% O_2_ and 5% CO_2_ balanced in N_2_, and (b) 21% O_2_ and 0.01% CO_2_ balanced in N_2_ and a 3-L syringe (Ultima CPX, MedGraphics, Saint Paul, MN, USA), respectively, following the manufacturer’s instructions. 

#### 2.4.3. Cerebral Activation 

Brain electrocortical activity was continuously assessed via electroencephalography (EEG) during experimental conditions. EEG activity was recorded by placing scalp surface electrodes at three brain locations known as F_4_, C_z_, and O_z_ [51]. The activity of the prefrontal, central motor, and occipital cortices was reflected in these cerebral areas, respectively. The adhered surface electrodes were attached to the scalp with a conductive electrode paste (Electron II, Conductivity Gel, Trenton, NJ, USA) to ensure the high quality of the EEG recording. The EEG signal was transmitted to the mobile data collection unit (TEL 100D, BIOPAC Systems, Inc., Goleta, CA, USA) from the recording electrodes (Ag-AgCl electrodes), then to the signal-processing system (MP 100A, BIOPAC Systems Inc., Goleta, CA, USA) and, in turn, to the computer (Acer, Aspire 5633 WLMi, Ljubljana, Slovenia). EEG signals were amplified, sampled at a frequency of 1000 Hz, and stored for further analysis. The EEG signal after collection was filtered (Band-pass filter 0.15–40 Hz) and analyzed in the 8–13 Hz (alpha band) and 13–30 Hz (beta band) frequency ranges (fast Fourier analysis), and subsequently the ratio a/b (a/b index) was calculated as the arousal level index (Acknowledge 7.3.3 Software, BIOPAC System Inc., Goleta, CA, USA). 

#### 2.4.4. Maximal Voluntary Contraction (MVC) and Electromyography (EMG)

Maximal isometric force and EMG activity for knee extensors was evaluated before and immediately after the end (~2.5 min) of the incremental exercise test. Briefly, participants were seated on a modified strength-training homemade chair, with their knee angle at 60°, hip angle at 90°, their trunk upright against the back of the chair, and their arms crossed over the chest and stabilized with Velcro straps. After a familiarization session consisting of five isometric muscle contractions of very low intensity, three high-intensity repetitions followed. After a 10 min rest, the maximal isometric knee extensor force was assessed by three 5-s MVCs of the right quadriceps with 1 min recovery between consecutive efforts. Participants were verbally encouraged to maximally activate their knee extensors throughout each exercise bout. The isometric force was recorded by an isometric dynamometer connected between the stationary chair and the participant’s ankle (SS-25, Biopac System Inc., Goleta, CA, USA). The highest force and EMG signal were recorded and used to normalize EMG activity during the incremental exercise test. 

EMG activity was recorded from the right vastus lateralis muscle using monitoring electrodes with full-surface adhesive hydrogel and a commercially available data acquisition system (MP 100A, EMG 100C, Biopac System Inc., Goleta, CA, USA) during the MVC assessment and the incremental exercise test to exhaustion. The recording electrodes were placed over the muscle belly of the vastus lateralis (i.e., at 2/3 of the way along the line between the anterior spinal iliac superior and the lateral side of the patella, in the direction of the muscle fiber) with an interelectrode distance of 20 mm and the reference electrode placed over an electrically neutral site, either on the patella of the same lower limb or the spina iliaca. Before electrode placement, the skin was carefully shaved and cleaned with alcohol. The EMG electrode location was marked with indelible marks to ensure that they were placed exactly in the same location at a subsequent visit. Electrodes and cables were strapped on the participants with medical hypoallergenic tape to minimize movement artifacts. 

The EMG signal was amplified by 1000 Hz, band-pass filtered (20–400 Hz), and sampled at 1 kHz with a 16-bit A/D converter (Acknowledge 7.3.3 software, Biopac System Inc., Goleta, CA, USA). The EMG signal amplitude was quantified as the calculation of the integrated EMG (iEMG; Acknowledge 7.3.3 software). The iEMG was determined over a ~0.5/1 sec around the maximal value for the MVC assessment for each muscle contraction over the last 15 sec of each incremental workload, and the values were then averaged. The iEMG values during exercise were expressed relative to the activity obtained during pre-exercise MVC trials (%MVC), and used as an indicator of muscle fiber recruitments during exercise [52].

#### 2.4.5. Perceptual Responses 

Rates of perceived discomfort (RPE) for dyspnea (RPE_dyspnea_) and leg fatigue (RPE_leg_) were quantified using Borg’s 6–20 scale [53] every 2 min throughout the experimental protocols. Furthermore, Borg’s CR-10 pain scale [54] was used to integrate any pain sensation due to the inflation of the thigh cuffs. 

#### 2.4.6. Data Analysis and Statistics 

Physiological responses were recorded continuously from the baseline until the end of the exercise. Rest values for all variables were averaged for the period of 6th to 9th minute during the 10 min resting period, either without or with muscle blood flow restriction. During incremental exercise, mean values over the last 15 s of each increment and the last 15 s at exhaustion were calculated for cardiopulmonary response, EMG, and EEG activity, as well as for muscle and cerebral oxygenation. Mean values of physiological responses were plotted against the relative peak power output (Rest and 20, 40, 60, 80, 90, and 100% PPO) during the incremental exercise for each experimental condition. A two-way ANOVA with repeated measures on both factors (condition × workloads) was used to evaluate the effects of the thigh cuffs’ application (NC and WC) across workloads (Rest and 20, 40, 60, 80, 90, and 100% PPO). Similarly, a comparison between the no cuff and with cuff conditions at absolute workloads of 60, 120, and 180 Watts was analyzed via a two-way ANOVA with repeated measures [condition (2 NC and WC) × workloads (60, 120, and 180 Watts)]. The significant main effects and interactions were then subjected to Tukey’s post hoc test. Values are reported as mean ± standard error of measurements (SEM), and two-tailed values of *p* < 0.05 were considered statistically significant.

## 3. Results

### 3.1. Exercise Tolerance 

All participants completed the incremental exercise test and reached maximal exertion in each exercise protocol. At exhaustion, participants were incapable of maintaining the appropriate rate of cycle pedaling, and RER values were greater than 1.10 in both conditions. RPE_leg_ was similar between the exercise protocols. There was a significant deterioration in exercise tolerance with muscle blood flow restriction: $\dot{V}$O_2max_ was reduced by 17 ± 2% (NC: 50.52 ± 2.16 mL·kg^−1^·min^−1^ vs. WC: 41.31 ± 1.87 mL·kg^−1^·min^−1^, *p* < 0.001), PPO by 28 ± 2% (NC: 324.92 ± 9.31 Watt vs. WC: 234.92 ± 7.64 Watt, *p* < 0.001), and time to exhaustion (TTE) by 28 ± 2% (NC: 654.41 ± 18.62 s vs. WC: 466.99 ± 14.71 s, *p* < 0.001). Thus, muscle blood flow restriction during exercise significantly limited exercise performance during cycling incremental exercise. 

### 3.2. Muscle Oxygenation 

Muscle blood flow restriction significantly altered resting skeletal muscle oxygenation. In detail, with muscle blood flow restriction, muscle-Δ[O_2_Hb], Δ[HHb], and Δ[THb] were significantly higher (*p* < 0.01), whereas muscle Δ[HbDiff] was significantly lower (*p* < 0.01) compared with the respective NC condition. During incremental exercise, there were significant main effects of “exercise condition”, “workload”, and “exercise condition × workload” interaction for muscle-Δ[O_2_Hb] and Δ[THb]. Specifically, muscle-Δ[O_2_Hb] gradually decreased from the beginning of exercise to 90% PPO and then remained low (Figure 2A), whereas muscle Δ[THb] was rapidly increased at the initiation of exercise and remained high until exercise termination (Figure 2C). The magnitude of muscle Δ[O_2_Hb] decline (Figure 2A) was significantly greater in the NC condition from 60–100% PPO (*p* < 0.001), while the muscle Δ[THb] increase (Figure 2C) was significantly greater in the WC condition from 20–100% PPO (*p* < 0.001). Furthermore, muscle-Δ[HHb] progressively increased (Figure 2B) and muscle Δ[HbDiff] decreased (Figure 2D) from the beginning of the test until the 80% PPO, and was then stabilized. The increase in muscle-Δ[HHb] and the decrease in muscle-Δ[HbDiff] were significantly greater from 20–40% PPO (*p* < 0.001) and at 20% PPO, (*p* < 0.001), respectively, in the WC compared to the NC condition. At absolute workloads, muscle-Δ[HHb] (*p* = 0.003) and Δ[THb] (*p* = 0.010) were significantly greater, whereas muscle-Δ[HbDiff] was significantly lower (*p* = 0.016) in the WC compared to the NC condition. No significant differences existed in muscle Δ[O_2_Hb] (*p* = 0.191) between the exercise protocols (Table 1). As expected, the indices of skeletal muscle oxygenation considerably declined during the incremental exercise to exhaustion; muscle blood flow restriction accelerated the rate of increase in muscle deoxygenation, and the decline in muscle oxygenation. 

### 3.3. Cerebral Oxygenation

At rest, muscle blood flow restriction did not result in significantly different values in cerebral-Δ[O_2_Hb], Δ[HHb], and Δ[THb] compared to the WC condition (all, *p* = 0.09–0.31), whereas cerebral Δ[HbDiff] was significantly lower (*p* = 0.03) in WC vs. the NC condition. During incremental exercise, there were a significant main effects of “exercise condition” (*p* = 0.023), “workload” (*p* < 0.001), and “exercise condition × workload” interaction (*p* < 0.001) for cerebral Δ[HHb] (Figure 3B). Specifically, cerebral-Δ[HHb] was unchanged at the initiation of exercise, and rapidly increased at 80% PPO and then stabilized until the end of exercise. The magnitude of cerebral Δ[HHb] increase was significantly greater in the NC condition compared to the WC condition from 80–100% PPO (*p* < 0.001), with no significant differences in exercise condition on cerebral-Δ[O_2_Hb], Δ[THb], and Δ[HbDiff] (all, *p* = 0.117–0.933). During incremental exercise, cerebral-Δ[O_2_Hb] increased continuously until the 60% PPO, and then stabilized until exhaustion (Figure 3A). The increase in Δ[O_2_Hb] was significantly greater in the NC condition compared to the WC condition (*p* < 0.001). Cerebral-Δ[THb] (Figure 3C) and Δ[HbDiff] (Figure 3D) were initially unchanged and rapidly increased at 80% PPO and 60% PPO, respectively, and then remained high until the end of exercise. The magnitude of Δ[THb] increase was significantly greater in the NC condition compared to the WC condition from 80–100% PPO (*p* < 0.001), whereas the increase in cerebral-Δ[HbDiff] was similar between exercise protocols (*p* > 0.05). No significant differences in cerebral Δ[O_2_Hb], Δ[HHb], Δ[THb], Δ[HbDiff] were observed during exercise at the absolute workload examined (60, 120, 180 W) between the NC and the WC condition (all *p* = 0.14–0.93; Table 1). Therefore, the cerebral oxygenation response to incremental exercise was similar between exercise protocols, whereas cerebral deoxygenation and total hemoglobin were significantly lower upon task failure with muscle blood flow restriction.

### 3.4. Cardiovascular Response

Resting cardiovascular responses (HR, SV, $\dot{Q}$, SBP, DBP, MAP, and TPR) were not different with muscle blood flow restriction compared to the NC condition (all, *p* = 0.81–1.00). There was a significant main effect of “exercise condition” on HR, SV, and $\dot{Q},$ with the mean values being significantly higher (*p* < 0.01) during exercise without muscle blood flow restriction compared to WC condition at all relative workloads (Figure 4A–C). During exercise, HR gradually increased (*p* < 0.001) from the start of exercise until exhaustion. Stroke volume increased during exercise until the 40% PPO and then stabilized. Thus, $\dot{Q}$ gradually increased during exercise until exhaustion, mainly due to an increase in HR and partially due to the increase in SV. Changes in HR, SV, and $\dot{Q}$ were significantly greater (*p* < 0.01) in the NC compared to the WC condition from 40–100% PPO. 

Regarding blood pressure responses, SBP, DBP, and MAP progressively increased (*p* < 0.01) from baseline until exhaustion (Figure 4D–F). The magnitude of the blood pressure increase was significantly greater in the WC compared to the NC condition (*p* < 0.05) at all relative workloads. No significant main effect of “exercise condition” (*p* = 0.07) on TPR was observed during exercise. However, TPR rapidly decreased at the initiation of exercise (20% PPO), and then stabilized until the end of exercise. At absolute workloads, HR was greater (*p* = 0.02), whereas SV (*p* = 0.0005) and $\dot{Q}$ (*p* = 0.0001) were lower in the WC compared to the NC condition. Blood pressure (SBP, DBP, MAP) and TPR were significantly greater (*p* < 0.001) during exercise at identical workloads in the WC compared to the NC condition (Table 1). These results indicate that blood flow restriction resulted in blunted hemodynamic responses at maximal exercise, despite the exaggerated blood pressure response. 

### 3.5. EEG Activity

The resting alpha, beta waves, and alpha-to-beta ratio of the EEG activity for all sites (F_4_, C_z_, and O_z_) were similar between NC and WC conditions (all, *p* = 0.23–0.96). Similarly, no significant main effect of “exercise condition” was observed for all sites of EEG activity during the incremental exercise test (all, *p* = 0.15–0.77). During exercise, only the alpha, beta waves, and alpha-to-beta ratio of Oz increased gradually compared to baseline (*p* < 0.01). No significant difference in EEG activity for the sites (F_4_, C_z_, and O_z_) was observed during exercise at absolute workloads between NC and WC conditions (all, *p* = 0.26–0.72). These data suggest that cerebral activation at exhaustion was identical between exercise protocols, despite the marked differences in $\dot{V}$O_2max_, PPO, and TTE.

### 3.6. EMG Activity and MVC Force

No significant main effect of “exercise condition” (*p* = 0.285) or “exercise condition × workload” interaction (*p* = 0.513) was observed for iEMG. The average iEMG over the incremental exercise test was 22.16 ± 2.66% of MVC and 16.17 ± 1.23% of MVC in the NC and WC conditions, respectively. Furthermore, iEMG increased progressively (*p* < 0.001) until the 80% PPO, and then remained high until the end of exercise (Figure 5). At any given absolute workload, iEMG was similar (*p* = 0.546) between the NC and WC conditions. Peak force during the 5-s MVC maneuvers significantly decreased from baseline after the $\dot{V}$O_2max_ trials (pre: 115.40 ± 9.44 kg vs. post: 113.45 ± 9.62 kg, *p* = 0.042), without differences in the magnitude of the reduction between conditions (NC: −2.61 ± 1.32% and WC: −3.23 ± 2.47%, *p* = 0.831). These results suggest that skeletal muscle activation and rate of motor unit recruitment were similar between exercise protocols during incremental exercise. Furthermore, skeletal muscle fatigue was similar between conditions, despite the marked differences in $\dot{V}$O_2max_, PPO, and TTE. 

### 3.7. Rate of Perceived Exertion 

The resting rate of perceived discomfort for the legs (RPE_leg_) was significantly higher (*p* < 0.001) with muscle blood flow restriction compared to the NC condition, whereas dyspnea (RPE_dyspnea_) did not differ (*p* = 0.33) between conditions. During exercise, RPE_leg_ and RPE_dyspnea_ increased gradually (*p* < 0.01) compared to baseline until the exercise was terminated. At exhaustion, RPE_dyspnea_ was significantly greater (*p* = 0.01) in NC (17 ± 0.4) than in the WC (15 ± 0.8) condition, whereas RPE_leg_ was similar between exercise protocols (NC: 18 ± 0.4 and WC: 19 ± 0.4, *p* = 0.392). At absolute workloads, RPE_leg_ and RPE_dyspnea_ were significantly greater (*p* < 0.001) in the WC compared to the NC condition in each workload (Table 1). Thus, perceptual leg discomfort in incremental exercise was significantly higher with muscle blood flow restriction, whereas at task failure, it was similar, suggesting that muscle blood flow restriction exaggerates the rate of increase in perceptual response. 

## 4. Discussion

The effect of restricting skeletal muscle oxygenation on hemodynamic responses, cerebral oxygenation, and cerebral activation during incremental exercise to exhaustion in male cyclists was investigated for the first time, both comprehensively and interactively. Furthermore, we elucidated whether the impairment of exercise performance with muscle blood flow restriction was associated with a premature decline in cerebral activation or/and with the inability of cardiac responses to reach their maximum. Confirming our original hypothesis, the results showed that skeletal muscle oxygenation restriction via the application of thigh cuffs reduced maximal aerobic capacity, as indicated by the decrease in $\dot{V}$O_2max_, in peak power output and in time to exhaustion. Furthermore, muscle blood flow restriction during whole-body dynamic exercise accelerated (a) the rate of skeletal muscle deoxygenation, (b) the rise in systolic blood pressure, (c) the rating of perceived exertion, and (d) the cerebral activation, without these variables being different at the point of exhaustion, even though $\dot{V}$O_2max_, peak power output, and time to exhaustion were significantly impaired in the WC compared to the NC condition. These novel data suggest that skeletal muscle oxygenation levels manipulated experimentally to a low level, in combination with an exaggerated blood pressure response and an earlier cerebral activation prematurely reaching its peak values, could be important determinants of exercise performance. Paradoxically, when arterial blood pressure reached its peak at exercise termination, the peak values of cerebral deoxygenation, regional cerebral blood volume, cardiac output, heart rate, and stroke volume were significantly compromised in muscle blood flow restriction compared with the control condition. 

### 4.1. Skeletal Muscle Oxygenation and Exercise Tolerance

As expected, the application of thigh cuffs at 120 mmHg during incremental exercise test to exhaustion resulted in significantly lower $\dot{V}$O_2max_ (17%), PPO (28%), and TTE (28%). This impairment of exercise capacity is consistent with that reported in other studies, with the average decrease ranging from 20–40%, depending on thigh cuff application [36] or/and lower body positive pressure [55,56,57].

Since 1924, researchers have been studying the limiting factors of $\dot{V}$O_2max_, and there is still debate over the critical factor. It seems that three functional entities play a decisive role either to a lesser or a greater extent in the cessation of exercise during incremental exercise to exhaustion: the brain, the heart, and the skeletal muscle. It is still unknown which of these three operating systems is the critical element that initiates the cessation of exercise [17,58,59,60,61]. We assumed that skeletal muscle oxygenation perturbation could modify either cerebral or/and cardiovascular activation, and that this might be a significant determinant of exercise intolerance with muscle blood flow restriction. 

### 4.2. Muscle Blood Flow Restriction and Skeletal Muscle Oxygenation during Exercise

Consistent with previous findings, muscle blood flow restriction exaggerated exercise-induced muscle Δ[HHb], Δ[THb], and Δ[O_2_Hb] changes during exercise at the same absolute and relative workloads [33,34,35,36,57] suggesting that thigh cuff application at 120 mmHg may be an effective method of reducing arterial blood flow to the muscles. At the same time, venous occlusion and blood pooling to lower extremities, as indicated by an increase in muscle Δ[HHb] and Δ[THb], resulted in a smaller venous return to the heart (as evidenced by the lower SV and $\dot{Q}$). It is worth mentioning that the muscle blood flow restriction intervention did not affect whole-body arterial oxygen availability (as indicated by finger pulse oxygen saturation (SpO_2_): 97 ± 0.4%) throughout the exercise.

A major finding of this study was that at task failure, the magnitude of exercise-induced skeletal muscle deoxygenation (Δ[HHb]) and oxygenation (Δ[HbDiff]) was similar between conditions, despite marked differences in performance time (~28%) and peak power output (~28%). Similar levels of skeletal muscle deoxygenation and oxygenation may indicate a similar level of muscle hypoxia in both conditions. These data suggest that muscle blood flow restriction accelerates the rate of muscle Δ[HHb] increases and Δ[HbDiff] reduction, and at the specific level at which these variables reach their maximal values, exercise terminates, regardless of the experimental condition. It seems that skeletal muscle oxygenation level could be a significant variable in determining exercise capacity during incremental exercise trials. 

Several studies that have investigated the limiting factors of $\dot{V}$O_2max_ through a wide range of experimental conditions (i.e., normoxia, hypoxia, iso-oxia, hyperoxia) have also reported a similar level of skeletal muscle oxygenation or/and deoxygenation at task failure, despite marked differences in arterial oxygen availability and exercise tolerance, without givingappropriate consideration [17,18,51,58,62,63,64,65]. Collectively, these findings could indicate that the rate of development of skeletal muscle deoxygenation or/and the rate of skeletal muscle oxygenation reduction until a ‘specific’ maximal level could be an important determinant of exercise tolerance. We interpret this hypothesis to indicate that the rate of skeletal muscle deoxygenation modification will affect exercise performance. In other words, accelerating skeletal muscle deoxygenation will impair exercise tolerance, whereas reducing skeletal muscle deoxygenation will improve exercise tolerance. However, at the point of exhaustion, the level of skeletal muscle deoxygenation or/and oxygenation will be identical.

It is well accepted that muscle blood flow restriction during dynamic exercise induces intramuscular metabolic perturbation. Indeed, muscle blood flow restriction has been reported to augment the build-up of metabolic by-products such as lactate (La), phosphocreatine (PCr), inorganic phosphate (Pi), and hydrogen ions (H^+^) compared to the control condition [66,67,68]. Furthermore, muscle blood flow reduction, mismatch of O_2_ delivery to metabolic demands, causes venous distention, and accumulate metabolic by-products activating group III/IV muscle afferents [69,70,71]. The activation of group III and IV muscle afferents appears to be essential for normal exercise hemodynamic and ventilatory responses, and at the same time contributes to the development of peripheral fatigue and facilitates central nervous system (CNS) fatigue by modulating motor cortical output and inhibiting motoneuronal output during high-intensity exercise [25,48,72,73,74,75]. 

Taken together, the inability to continue high-intensity incremental exercise appears to coincide with a specific low level (limit) of skeletal muscle oxygenation. It seems that a skeletal muscle oxygenation or/and deoxygenation limit could be the initial stimulus that triggers exercise cessation via modification of either cerebral activation (central nervous system) or/and cardiac response. 

### 4.3. Skeletal Muscle Oxygenation and Cerebral Oxygenation during Exercise

In the present study, cerebral oxygenation indices were gradually modified in response to incremental exercise, indicating a progressive increase in cerebral activation and metabolism, as other studies have shown [42,58,62,64,76,77]. To the best of our knowledge, this is the first study to evaluate the effects of muscle blood flow restriction during incremental exercise to exhaustion on cerebral oxygenation. In a recent study by our laboratory, restricting skeletal muscle oxygenation via thigh cuff application (120 mmHg) at rest resulted in alterations in cerebral hemodynamics [36]. In this study, we found that the magnitude of changes in cerebral Δ[HHb] and Δ[THb] was significantly lower in the WC compared to the NC condition, while no significant differences were observed in cerebral Δ[O_2_Hb] and Δ[HbDiff]) at the point of exhaustion. These results were accompanied by a lower increase in metabolic rate, cardiac output, and a greater increase in mean arterial pressure. Indeed, $\dot{V}$O_2_, $\dot{V}$CO_2_, and $\dot{Q}$ were significantly lower, and MAP was significantly higher at task failure with muscle blood flow restriction. The lower regional cerebral blood volume in the WC condition is possibly associated with the lower SV_max_ and HR_max_ during the thigh cuffs’ application. 

Although cerebral perfusion and oxygenation impairments have been demonstrated to be a limiting factor in exercise performance during incremental exercise to exhaustion, due to inadequate motor unit recruitment causing premature fatigue [77], this was not the case in the present study. Indeed, the magnitude of changes in cerebral [O_2_Hb], [HHb], [THb], and [HbDiff] reported in the present study without and with muscle blood flow restriction at task failure was not detrimental, contrary to the results of other studies [17,18,42,59,77,78]. Furthermore, the similar cerebral oxygenation index (Δ[HbDiff]) at the point of exhaustion suggested a balance between cerebral oxygen delivery and oxygen utilization regardless of muscular disturbance and power output performed. Additionally, the rate of motor unit recruitment (iEMG) and the magnitude of peripheral muscle fatigue in our study were similar during incremental exercise between exercise protocols, despite marked differences in exercise capacity. Thus, there is no evidence that muscle blood flow restriction-induced crucial cerebral hemodynamic changes cause premature fatigue and explain exercise intolerance. 

### 4.4. Skeletal Muscle Oxygenation and Cerebral Activation during Exercise

Cerebral electrocortical activity, muscular electrical activity, and rating of perceived exertion are considered indexes of central command. In an innovative approach, all these factors were examined in the present study, and it was found that they were gradually increased during incremental exercise to exhaustion, indicating a progressive cerebral activation. The magnitude of EEG and EMG increase was not affected by muscle blood flow restriction at the same absolute and relative workload, and as a consequence, the cerebral and skeletal muscle activation at task failure were supposed to be maximal and identical between experimental conditions, despite marked differences in TTE, PPO and $\dot{V}$O_2max_. In other words, as with cerebral hemodynamic responses, exercise terminates as far as central command reaches its maximal capacity, regardless of the power output produced. 

Furthermore, perception of exertion in dynamic exercise has been considered a combined effect of central activation and muscle afferent feedback [79,80]. In our study, the rate of leg fatigue perceived exertion (RPE_leg_) was higher in the WC condition, even at rest, progressively increased during incremental exercise in both conditions, and was greater with muscle blood flow restriction at a given workload. However, RPE_leg_ was similar between experimental conditions at task failure, despite large differences in PPO, TTE, and $\dot{V}$O_2max_. It has been suggested that muscle blood flow restriction accelerated the rate of increase in RPE during exercise, which is also an indication of increased cerebral activation. It is worth mentioning that the rate of increase in RPE during dynamic exercise has been shown to predict maximal aerobic power and the time to exhaustion [62,63]. In the current study, the rate of RPE_leg_ increase was highly and inversely correlated with $\dot{V}$O_2max_ (r = −0.70, *p* < 0.0001), PPO (r = −0.79, *p* < 0.0001), and TTE (r = −0.81, *p* < 0.0001). It seems that the higher rate of RPE_leg_ increase during exercise with the application of thigh cuffs could be an indication that the brain senses (probably via muscle afferent input) the different muscle oxygenation levels. The above assumption is supported by the existence of a significant correlation between skeletal muscle oxygenation levels and RPE values (r = 0.45, *p* = 0.02). Collectively, these data suggest that muscle blood flow restriction during incremental exercise to exhaustion augments the activation of the central nervous system, and as a result, accelerates the rate of increase in EMG, EEG activity, and RPE, until a critical level of skeletal muscle deoxygenation and muscle oxygenation is achieved (i.e., the participants were no longer able to continue the exercise). The identical levels of skeletal muscle activity, cerebral activation, and perceptual response at task failure, despite marked differences in exercise performance, may constitute limiting factors, but not factors explaining system collapse, determining exercise performance and revealing power output differences.

### 4.5. Skeletal Muscle Oxygenation and Cardiovascular Response during Exercise 

In the present study, muscle blood flow restriction during exercise significantly affected cardiac responses. Indeed, HR was significantly increased, whereas SV and $\dot{Q}$ were reduced during exercise at the same absolute workload with muscle blood flow restriction. Furthermore, cardiac responses were significantly impaired at task failure with muscle blood flow restriction. In fact, HR_max_, SV_max_, and $\dot{Q}$_max_ were significantly lower by 9%, 18%, and 14%, respectively, in the WC compared to the NC condition. It seems that maximal cardiac responses were also restricted with muscle blood flow restriction, and exhaustion was achieved faster, with a lower PPO and $\dot{V}$O_2max_. 

Previous studies have reported that cardiovascular responses at task failure were either compromised [36,69] or unaffected [56,57,81,82] when external pressure was applied to the thighs, despite marked differences in exercise tolerance being recorded. Some of the discrepancies may be attributed to methodological issues such as the restrictive occlusion pressure applied (90–200 mmHg), the method of reducing blood flow (pressure cuffs, lower body positive or negative pressure), and the body positions during exercise (seating vs. supine). 

Maximum cardiovascular response and the oxygen delivery system are widely accepted as some of the major limiting factors of $\dot{V}$O_2max_ and aerobic exercise capacity, according to the cardiovascular/anaerobic model of exercise performance [26,27,28]. Specifically, exercise capacity is limited by cardiac function, and exhaustion is exhibited after insufficient oxygen delivery to the active skeletal muscle. In this study, cardiac output was significantly reduced during incremental exercise with muscle blood flow restriction. We observed that lower cardiac output was mainly due to a lower stroke volume, despite higher HR values when pressure was applied to the thigh at the same absolute workloads. At exhaustion, however, maximal cardiac output was severely compromised due to both a lower stroke volume and heart rate. It seems that the maximum pumping capacity of the heart remained submaximal at task failure with muscle blood flow restriction, compared to the control condition, which may be a significant factor in exercise intolerance. 

Concerning limited stroke volume during exercise with muscle blood flow restriction, venous occlusion has been reported [33,35,36,57] to induce arterial inflow reduction and blood pooling in capacitance vessels distal to the cuff in a pressure-dependent manner, which compromises both venous return to the heart and stroke volume output. The above inference is further supported by the negative significant correlation, observed in the present study, between SV and muscle Δ[THb] (r = −0.48, *p* < 0.01). 

It appears that the reduced HR_max_ with the application of thigh cuffs seems to contribute to the reduction in $\dot{Q}$_max_ and the subsequent exercise intolerance. However, the underlying mechanisms are still unknown. Reduced HR_max_ may simply be the result of a lower maximum workload. It is well known that there is a close linear relationship between HR and workload from rest to maximal exercise. However, several studies that manipulated the heart rate response to incremental exercise to exhaustion through acute hypoxia and hyperoxia, chronic systemic hypoxia, and sympathetic or/and parasympathetic blockade reported that maximal heart rate does not always follow the $\dot{V}$O_2_ and workload fluctuations [12,83,84,85]. In our study, we observed that restricted skeletal muscle oxygenation had no effect on HR at rest, increased HR during exercise at the same absolute workload, and significantly reduced HR at exhaustion. In addition, we found no significant correlation between HR_max_ and PPO in the NC (r = 0.153, *p* = 0.617) and WC (r = 0.449, *p* = 0.124) conditions. It seems that the reduced HR_max_ was not caused simply by the lower power output produced. 

Muscle blood flow restriction during exercise was associated with substantial sympathoexcitation (exercise pressor reflex) due to venous occlusion and the accumulation of metabolic by-products [70,71] to facilitate adequate muscle perfusion, thus restoring a mismatch between muscle metabolic demands and oxygen delivery [24,86,87,88]. Therefore, along with the blood pressure increase, we expected at least HR_max_ to be attained during exercise with muscle blood flow restriction, but this expectation was never fulfilled. In brief, a compromised maximal heart rate under conditions of muscular hypoxemia is not caused by less work done, and this fact constitutes a paradox, considering that in such conditions, full sympathetic stimulation exists. 

It should be noted that SBP was similar and DBP was higher at exhaustion between experimental protocols, despite the marked differences in PPO and time to exhaustion. In this study, the augmented blood pressure response during dynamic exercise with thigh cuff application is able to activate arterial baroreceptors (i.e., loaded), and consequently, arterial baroreflex acts to suppress the rising arterial blood pressure [89,90]. Furthermore, arterial baroreflex activation during high-intensity dynamic exercise influences muscle metaboreflex-mediated cardiac acceleration and sympathoexcitation, and hence it is assumed that exercise will stop with a similar arterial blood pressure and a lower heart rate and cardiac output. It remains to be established, via experimentation, whether the arterial baroreflex is modulated during incremental exercise to exhaustion with muscle metaboreflex activation, and whether this reflexively sets the limits of heart rate (bradycardia), cardiac output, and blood pressure, thereby leading to termination of exercise.

A special methodological issue is that venous occlusion and blood pooling to the lower extremities affects a wide range of physiological responses, consequently producing different exercise conditions from those without muscle blood flow restriction. Thus, these results may not necessarily apply under physiological exercise conditions. However, we used the muscle blood flow restriction intervention during exercise to restrict skeletal muscle oxygenation, and thus to stress the O_2_ delivery system, in order to investigate the possible determinants of exercise tolerance. It is worth mentioning that the muscle blood flow restriction method appears to induce physiological responses similar to those observed in chronic diseases such as hypertension, heart failure, and peripheral vascular disease [23,91,92]. Thus, muscle blood flow restriction intervention could be used in the future to investigate the development of cardiovascular diseases. Furthermore, the cuff occlusion pressure was set at 120 mmHg for all participants, and was not individualized according to the individual arterial occlusion pressure. An absolute cuff occlusion pressure value of 120 mmHg has been shown to elicit blood flow restriction of approximately 55–65% [50]. These blood flow reductions are widely used in resistance and aerobic exercise intervention with muscle blood flow restriction to enhance skeletal muscle and aerobic adaptations in health and diseases [29,30,31,32]. Another limitation of this study was that it included only young healthy male athletes with high physical fitness. Thus, these results may not necessarily apply to individuals with different characteristics. Future studies should include nonathletes, females, and patients with chronic diseases. 

## 5. Conclusions

In conclusion, limiting skeletal muscle oxygenation resulted in a deterioration of maximal aerobic capacity during incremental exercise to exhaustion. Despite the marked decrease in exercise performance ($\dot{V}$O_2max_, peak power output, and time to exhaustion), skeletal muscle deoxygenation, systolic blood pressure, rating of perceived exertion, and cerebral activation at exhaustion were identical in both experimental conditions. The aforementioned factors reached prematurely a ‘critical’ maximal response with muscle blood flow restriction. The impaired exercise tolerance was associated with deterioration in cardiac responses (HR_max_, SV_max_, $\dot{Q}$_max_). This study was the first one which examined in a comprehensive and integrative manner how the reduction of skeletal muscle oxygenation via thigh cuff application would simultaneously affect cerebral and cardiovascular responses, during an incremental exercise to exhaustion protocol, in male cyclists. It was proved that limiting skeletal muscle oxygenation augments cerebral activation too early, and does not allow peripheral cardiovascular response to reach its maximal capacity. As a consequence, maximal oxygen uptake is limited. Thus, our findings highlight the importance of skeletal muscle oxygenation as a possible determinant of exercise tolerance during maximal whole-body exercise, as it accelerates cerebral activation reaching its peak, and impairs maximal cardiac responses. Further research is required to elucidate the causal link between cerebral and cardiac cross-communication during blood flow restriction. Furthermore, our results provide considerable insight into the physiological responses during incremental exercise to exhaustion with muscle blood flow restriction in young athletes, in whom this intervention is often used to enhance exercise performance. Finally, exercise intolerance in conditions such as chronic heart failure, hypertension, peripheral vascular diseases, and vascular claudication, which are characterized by skeletal muscle perfusion impairments, compromised oxygenation, and blood pooling, can be considered through the simulation of muscle blood flow restriction in healthy individuals. In recent years, athletes of various fitness levels have used this method (muscle blood flow restriction) at rest (known as ischemic preconditioning) and during resistance and aerobic exercise training protocols as an alternative intervention to improve exercise performance and enhance skeletal muscle and aerobic adaptations.

## Figures and Tables

**Figure 1 biology-12-00981-f001:**
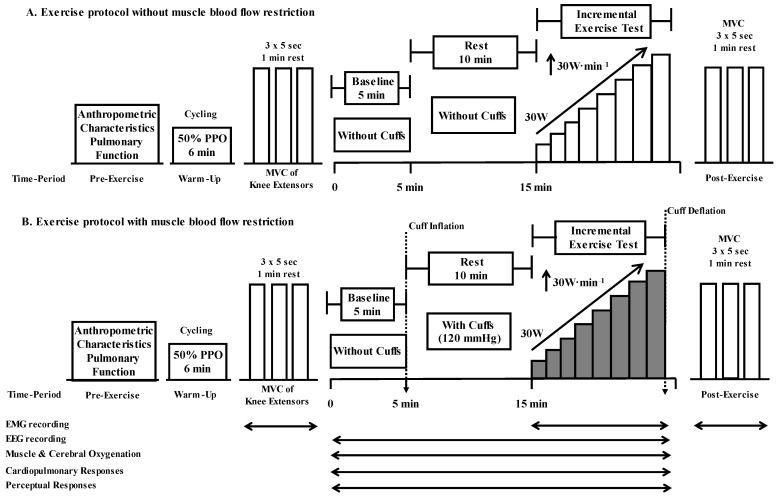
Overview of the experimental protocol, which consisted of an incremental exercise test to exhaustion on a cycle ergometer on two separate conditions: (**A**) without (no cuffs, NC) and (**B**) with (with cuffs, WC) muscle blood flow restriction. Brain electroencephalography activity (EEG), skeletal muscle activity (EMG), muscle oxygenation, cerebral oxygenation, cardiopulmonary, and perceptual responses were measured during the exercise protocol. Maximal voluntary isometric contraction (MVC) of the knee extensors was evaluated before and immediately after the end (~2.5 min) of the incremental exercise test. PPO: peak power output.

**Figure 2 biology-12-00981-f002:**
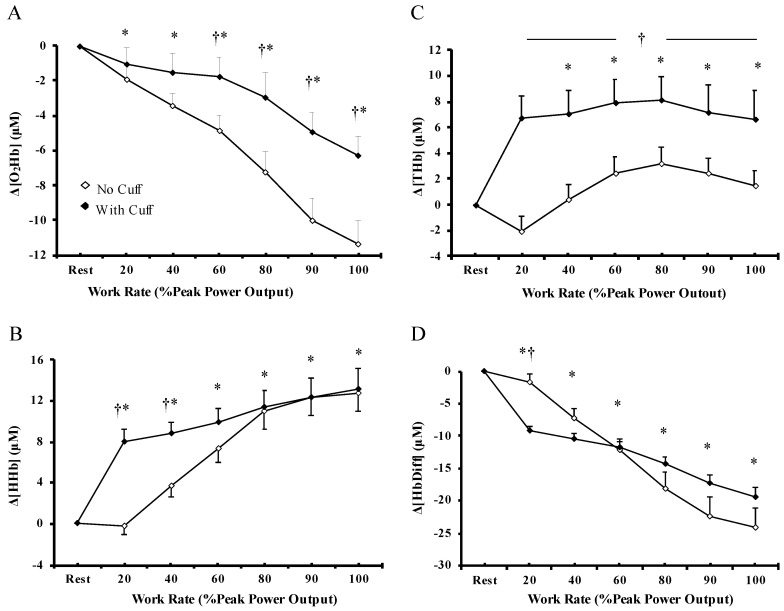
Muscle concentration changes in (**A**) oxyhemoglobin, (**B**) deoxyhemoglobin, (**C**) total hemoglobin, and (**D**) hemoglobin difference during incremental cycle ergometer test without (◊) and with (♦) muscle blood flow restriction at relative work rate. Values are means ± SEM of 13 participants. † Significant differences between no cuff and with cuff condition (*p* < 0.05). * Significant differences compared to rest (*p* < 0.01).

**Figure 3 biology-12-00981-f003:**
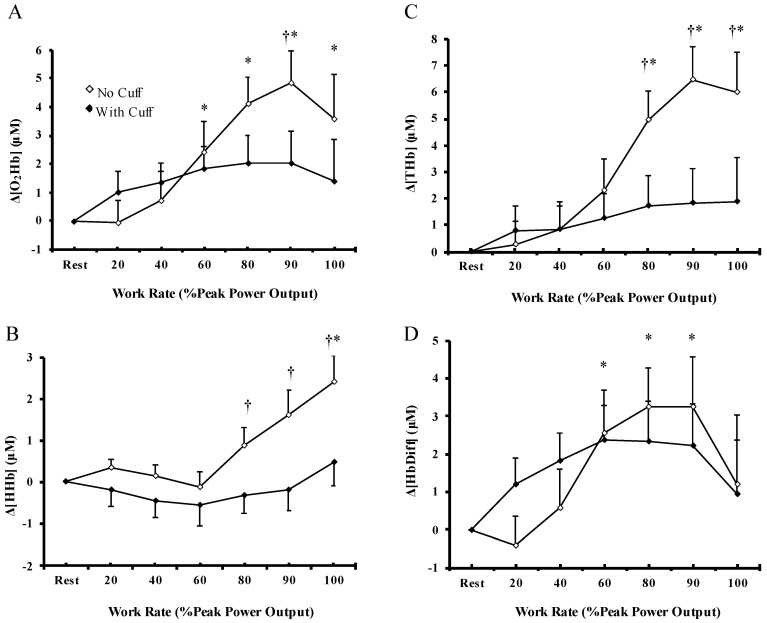
Cerebral concentration changes in (**A**) oxyhemoglobin, (**B**) deoxyhemoglobin, (**C**) total hemoglobin, and (**D**) hemoglobin difference during incremental cycle ergometer test without (◊) and with (♦) muscle blood flow restriction at relative work rate. Values are means ± SEM of 13 participants. † Significant differences between no cuff and with cuff condition (*p* < 0.05). * Significant differences compared to rest (*p* < 0.01).

**Figure 4 biology-12-00981-f004:**
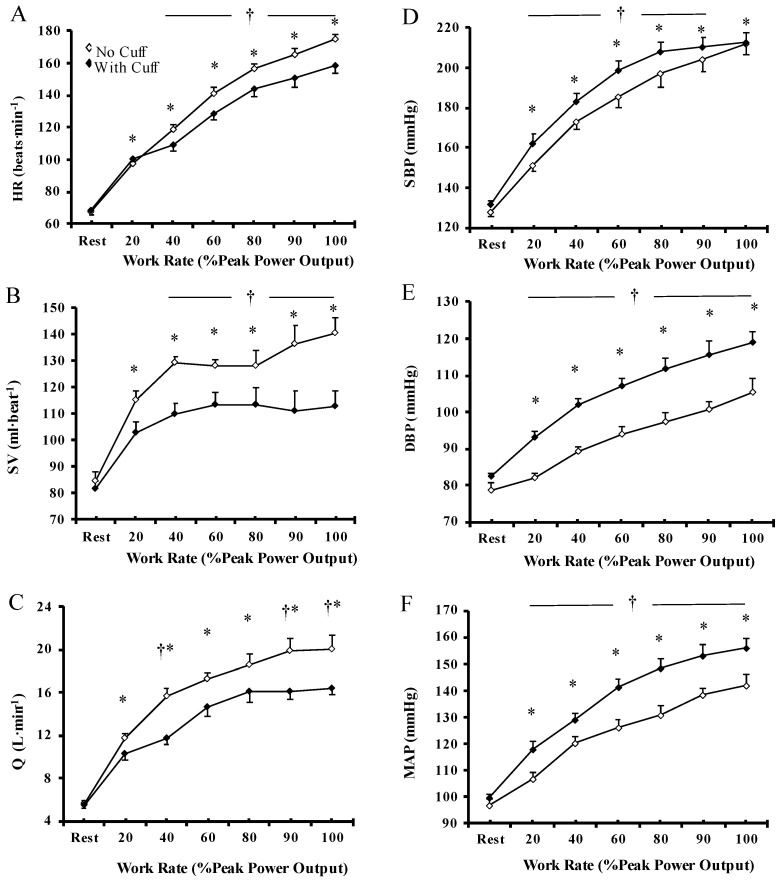
Cardiovascular response during incremental cycle ergometer test without (◊) and with (♦) muscle blood flow restriction. Heart rate (HR; (**A**)), stroke volume (SV; (**B**)), cardiac output ($\dot{Q}$; (**C**)), systolic blood pressure (SBP; (**D**)), diastolic blood pressure (DBP; (**E**)), and mean arterial pressure (MAP; (**F**)). Values are means ± SEM of 13 participants. † Significant differences between no cuff and with cuff condition (*p* < 0.05). * Significant differences compared to rest (*p* < 0.01).

**Figure 5 biology-12-00981-f005:**
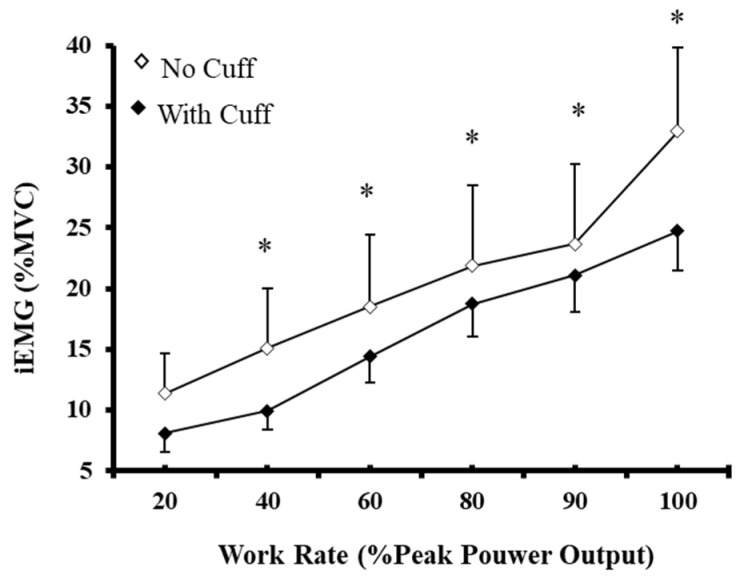
Integrated electromyography (iEMG) activity during the incremental cycle ergometer test without (◊) and with (♦) muscle blood flow restriction. Values are means ± SEM of 13 participants. * Significant differences compared to 20% PPO (*p* < 0.01).

**Table 1 biology-12-00981-t001:** Mean values ± SEM (*n* = 13) of physiological responses during the incremental exercise test with an identical workload without (no cuff, NC) and with (with cuff, WC) muscle blood flow restriction.

	No Cuff Condition			With Cuff Condition		
Variables	60 W	120 W	180 W	60 W	120 W	180 W
Muscle oxygenation						
Δ[O_2_Hb]m (μΜ)	−1.76 ± 0.74	−3.20 ± 0.70	−4.48 ± 0.92 *	−1.06 ± 0.99	−1.12 ± 1.10	−2.83 ± 1.19 *
Δ[HHb]m (μΜ)	0.06 ± 0.92	3.46 ± 1.22 *	6.71 ± 1.46 *	8.10 ± 1.14 ^†^	8.64 ± 1.42 ^†^	11.14 ± 1.63 ^†,^*
Δ[THb]m (μΜ)	−1.69 ± 1.16	0.26 ± 1.24 *	2.22 ± 1.26 *	6.77 ± 1.90 ^†^	7.25 ± 1.96 ^†^	8.03 ± 2.00 ^†^
Δ[HbDiff]m (μΜ)	−1.82 ± 1.21	−6.67 ± 1.54 *	−11.19 ± 2.09 *	−9.15 ± 1.10 ^†^	−9.76 ± 1.68 ^†^	−13.97 ± 2.05 ^†,^*
Cerebral oxygenation						
Δ[O_2_Hb]c (μΜ)	−0.04 ± 0.68	0.61 ± 0.85	1.90 ± 0.93	0.73 ± 0.79	1.54 ± 0.91	1.87 ± 0.97
Δ[HHb]c (μΜ)	0.40 ± 0.25	0.25 ± 0.30	−0.22 ± 0.30	−0.12 ± 0.31	−0.55 ± 0.37	−0.28 ± 0.43
Δ[THb]c (μΜ)	0.36 ± 0.78	0.86 ± 0.90	1.68 ± 0.99	0.61 ± 0.91	1.00 ± 1.00	1.59 ± 1.10
Δ[HbDiff]c (μΜ)	−0.45 ± 0.67	0.36 ± 0.91	2.12 ± 0.97	0.85 ± 0.78	2.09 ± 0.96	2.15 ± 1.03
Hemodynamics						
HR (beats·min^−1^)	97 ± 3	116 ± 3 *	137 ± 3 *	101 ± 3 ^†^	123 ± 4 ^†,^*	141 ± 4 ^†,^*
SV (ml·beat^−1^)	115.08 ± 3.32	124.65 ± 2.49 *	127.98 ± 2.77	104.17 ± 4.44 ^†^	109.65 ± 4.98 ^†,^*	112.44 ± 3.75 ^†^
$\dot{Q}$ (L·min^−1^)	11.71 ± 0.42	14.54 ± 0.56 *	17.35 ± 0.73	10.59 ± 0.63 ^†^	13.34 ± 0.88 ^†,^*	15.34 ± 0.72 ^†^
SBP (mmHg)	152± 3	164 ± 5 *	186 ± 5 *	169 ± 4 ^†^	193 ± 4 ^†,^*	206 ± 5 ^†,^*
DBP (mmHg)	82 ± 1	86 ± 2	95 ± 1 *	95 ± 1 ^†^	105 ± 2 ^†,^*	111 ± 2 ^†,^*
MAP (mmHg)	107 ± 2	113 ± 4 *	127 ± 2 *	121 ± 2 ^†^	138 ± 3 ^†,^*	148 ± 3 ^†,^*
TPR (mu)	0.57 ± 0.02	0.52 ± 0.03 *	0.51 ± 0.03	0.74 ± 0.05 ^†^	0.67 ± 0.05 ^†,^*	0.61 ± 0.03 ^†,^*
Exercise capacity/ Metabolic parameters						
$\dot{V}$O_2_ (ml·min^−1^)	1104 ± 44	1802 ± 64 *	2419 ± 63 *	1186 ± 45 ^†^	1962 ± 57 ^†,^*	2602 ± 63 ^†,^*
$\dot{V}$CO_2_ (ml·min^−1^)	887 ± 45	1526 ± 66 *	2280 ± 79 *	940 ± 42 ^†^	1742 ± 60 ^†,^*	2624 ± 89 ^†,^*
$\dot{V}$_E_ (L·min^−1^)	27.23 ± 1.31	41.54 ± 1.63 *	58.07 ± 2.20 *	28.77 ± 1.44	47.76 ± 1.75 ^†,^*	72.74 ± 4.23 ^†,^*
RER	0.80 ± 0.02	0.85 ± 0.02 *	0.95 ± 0.02 *	0.79 ± 0.02	0.89 ± 0.02 *	1.01 ± 0.03 ^†,^*
V_T_ (ml·breath^−1^)	1259 ± 52	1730 ± 59 *	2204 ± 82 *	1252 ± 55	1884 ± 96 *	2311 ± 144 *
B_f_ (breaths·min^−1^)	22 ± 1	24 ± 1	27 ± 1 *	24 ± 2	26 ± 2	33 ± 3 ^†,^*
Cerebral activation						
iEMG (%MVC)	11.38 ± 3.32	16.94 ± 4.87 *	20.33 ± 5.91 *	8.65 ± 1.39	13.49 ± 1.91 *	18.04 ± 2.40 *
Fatigue perception						
RPE_dyspnea_	6.38 ± 0.14	8.92 ± 0.43 *	10.77 ± 0.44 *	6.77 ± 0.32 ^†^	10.23 ± 0.46 ^†,^*	12.23 ± 0.75 ^†,^*
RPE_leg_	8.08 ± 0.33	10.00 ± 0.67 *	12.69 ± 0.33 *	11.82 ± 0.52 ^†^	14.33 ± 0.38 ^†,^*	16.46 ± 0.54 ^†,^*

Δ[O_2_Hb]m, Δ[HHb]m, Δ[THb]m and Δ[HbDiff]m: oxy-, deoxy-, total and hemoglobin difference in muscle tissue, respectively; Δ[O_2_Hb]c, Δ[HHb]c, Δ[THb]c and Δ[HbDiff]c oxy-, deoxy-, total and hemoglobin difference in cerebral tissue, respectively; HR: heart rate; SV: stroke volume; $\dot{Q}$: cardiac output; SBP: systolic blood pressure; DBP: diastolic blood pressure; MAP: mean arterial pressure; TPR: total peripheral resistance; $\dot{V}$O_2_: oxygen consumption; $\dot{V}$CO_2_: carbon dioxide production; $\dot{V}$_E_: ventilation; V_T_: tidal volume; B_f_: breath frequency; P_ET_O_2_: end-tidal partial pressure of oxygen; P_ET_CO_2_: end-tidal partial pressure of carbon dioxide; iEMG: integrated electromyography; RPE_dyspnea_: rate of perceived exertion for dyspnea; RPE_leg_: rate of perceived exertion for leg fatigue. ^†^ Significant differences between no cuff and with cuff condition (*p* < 0.05). * Significant differences compared to the previous workload (*p* < 0.01).

## Data Availability

The data are not publicly available due to privacy.

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
