# Peer review of "The Effect of Skeletal Muscle Oxygenation on Hemodynamics, Cerebral Oxygenation and Activation, and Exercise Performance during Incremental Exercise to Exhaustion in Male Cyclists"

_biology, 2023, doi:10.3390/biology12070981_

Round 1
Reviewer 1 Report (Previous Reviewer 1)
To authors
This study examined the effect of local muscle occlusion on exercise performance with simultaneous measurement of oxygen-associated parameters in brain, heart, and skeletal muscle. And authors concluded that local occlusion-induced exercise intolerance is associated with an enhanced cerebral activation (?) and a blunted cardiovascular responses. Experimental method itself is simple, and a large part of methods are standard techniques, so results provided seems reliable. In contrast, present purpose seems not clear-cut or very weak. At least, authors seemed to have failed to highlight their originality and its importance in the presentation.
Please see comments below.
Major
It is very hard to follow the presentation. In the present study, authors kindly provided several results, but it might make presentation difficult to understand. Basically, outstanding results in the present study seems not much. I think sentence in line 111-113 might be main purpose and key point. Author might want to re-arrange (basically shorten) your presentation along with go-straight line to your conclusion.
In line 29, authors mentioned that the present purpose is to explore the limiting factors of aerobic capacity. This purpose is too abstractive. In association with this, please make sure whether sentence in line 41-44 is adequate answer to your purpose.
Unfortunately, I did not understand well about results of the “enhanced cerebral activation” as mentioned in line 43. In my knowledge, authors said in previous draft like “unaltered central nervous system…”. Please make sure of your results and interpretation.
Minor
In the present study, authors found that muscle local occlusion resulted in lower muscle oxygenation level, increased blood pressure, increased perceived exertion level, increased cerebral activation level (?) and lower cardiovascular responses as described in simple summary. However, in line 21, authors deleted cardiovascular responses as important determinants, but again appeared in line 43 in abstract. Is it important result or not? Does cardiovascular system include blood pressure or not? Does “central nervous system” (line 21) include cerebral activation (line 43)? How about HR? is this a part of central nervous system (under autonomic nerve)? Please categorize properly. It is very difficult to follow authors’ intention.
Line 24. “This experimental approach could be used in the investigation of the mechanisms of cardiovascular dysfunction during exercise.”
Reviewer think this sentence seems difficult to be understood by journal readers if readers read only “simple summary”. What does “dysfunction” mean? exhausted condition in exercise as normal regulation?? If authors want to try picking up the usefulness of your model, apparently explanation is deficient in simple summary. Please again re-consider your point to be emphasized here.
In line 59, authors divided factors into central vs. peripheral. Please make sure again whether a) to d) factor is “central” factor or not.
In line 90, “paradoxically”?
Ref 36 and 37 are published by your group. Please clearly mention it in the text.
Line 125: “for the first time in a holistic and integrative approach” seems abstract expression. Please have re-consideration.
Line 148: “Participants visited the laboratory three times to become familiar with the experimental procedures” means total 3 visiting (practice, task1, task2) or total 5 visiting (3-times practice, task1, task2)?
Line 216: “Muscle and cerebral measurements were normalized to zero before recording baseline values to express the magnitude of muscle and cerebral deoxygenation at rest and maximal exercise without and with muscle blood flow restriction.”
What does “baseline” mean? When did you measure it? before occlusion or after occlusion? Please mention it clearly.
If “after occlusion”, why did you not use the before-occlusion value as zero point. Please give me your comment.
In line 329, authors mentioned “Specifically, muscle Δ[O2Hb] gradually decreased from the beginning of exercise to 90%PPO and then remained low (Figure 1, A1, B1)”. However, X axis of A1 is not %PPO. Also, at the 100%, I can see further reduction after 90%. Please make sure.
Line 348: n=13, not n-13.
Line 333: The magnitude of muscle Δ[O2Hb] decline was significantly greater in NC condition from 60-100%PPO (p<0.001), while the muscle Δ[THb] rise was significantly greater in the WC condition from 20-100%PPO (p<0.001).” Authors may be talking about fig B1 for the Δ[O2Hb], is the right? Which fig is for THb? By any chance, are you talking about Δ[HHb], not THb? Because I can see†symbol on 20-100%PPO in fig B2, not in B3. Please make sure and demonstrate clearly.
Line 373: “Specifically, cerebral-Δ[HHb] was unchanged at the initiation of exercise and rapidly increased at 90%PPO and then stabilized until the end of exercise.”
90%?, not 80%? in fig 2B2.
In the present study, authors used specific model of local occlusion to overcome experimental problem obtained in environmental (e.g., hypoxic) experiment. In line 80-88, authors have mentioned usefulness of experimental model. At now, after present experiment, how should we think advantage of local occlusion model? is it possible to apply present specific results to physiology appeared under regular (normal) exercise? What is the difference among exercise-induced fatigue, exercise- and hypoxia-induced fatigue, and occlusion-induced fatigue? Please give us your summary about experimental model itself. This may contribute to increase scientific value of present study.
In line 42, authors only mentioned “association” as parameters moving together under local occlusion. In contrast, line 21, it seems difficult to say whether these changes are important (critical) or not because authors just see movement of parameters. Please tune the tone of your statement/expression.
Is there any relation between cerebral oxygenation level and cerebral EEG level? How does EEG move under environmental (e.g.,hypoxia) challenge? Is this different from your experimental model? Please add information in the text.
In line 106, suddenly “cerebral activation” and “oxygenation” came up to introduction as factors for fatigue/vo2max-limiting factor. How does it affect on fatigue? Please add additional brief explanation at first appearance.
In figures, literature size seems very small. Would you kindly re-arrange figures?
In the present study, results are expressed in relative and absolute manner, at some sampling point (baseline, rest, during exercise, exhaustion) with comparison to apposite group, previous sampling point. So, it is difficult to follow and understand results. Especially, concept/interpretation against during exercise and at exhaustion, and against relative and absolute expression seems have some difficulty in adequate understanding. Reviewer requests additional arrangement/ ingenuity to your presentation.
Author Response
REVIEWER'S COMMENTS to Author:
Reviewer: 1
Reviewer’s comment
Comments and Suggestions for Authors
To authors:
This study examined the effect of local muscle occlusion on exercise performance with simultaneous measurement of oxygen-associated parameters in brain, heart, and skeletal muscle. And authors concluded that local occlusion-induced exercise intolerance is associated with an enhanced cerebral activation (?) and a blunted cardiovascular responses. Experimental method itself is simple, and a large part of methods are standard techniques, so results provided seems reliable. In contrast, present purpose seems not clear-cut or very weak. At least, authors seemed to have failed to highlight their originality and its importance in the presentation.
Please see comments below
Response:
Major:
Reviewer’s comment
It is very hard to follow the presentation. In the present study, authors kindly provided several results, but it might make presentation difficult to understand. Basically, outstanding results in the present study seems not much. I think sentence in lines 111-113 might be main purpose and key point. Author might want to re-arrange (basically shorten) your presentation along with go-straight line to your conclusion.
Response:
We thank the reviewer for their comment. Following the reviewer’s suggestion, we have only retained the results that are directly related to the main purpose of this study. The main purpose of the current study was to elucidate whether exercise intolerance with muscle blood flow restriction was located at central nervous system or at the cardiac response associated with increased central nervous system activation or/and deterioration in cardiovascular response. Furthermore, we have attempted to shorten the manuscript’s length (please see the revised manuscript).
Reviewer’s comment
In line 29, the authors mentioned that the present purpose is to explore the limiting factors of aerobic capacity. This purpose is too abstractive. In association with this, please make sure whether sentence in lines 41-44 is an adequate answer to your purpose.
Response:
We thank the reviewer for their comment. We would like to mention that we have rephrased the purpose of the present study in the abstract section of the revised manuscript (page 1, lines 26-28) to specify it which now reads as follows ‘This study aimed to elucidate whether muscle blood flow restriction during maximal exercise was associated with alterations in hemodynamics, cerebral oxygenation, cerebral activation, and de-terioration of exercise performance in males.’
Furthermore, following the reviewer’s suggestion, we have modified the conclusion sentences in the abstract section of the revised manuscript (pages 1-2, lines 42-46) which now reads as follows ‘In conclusion, in trained cyclists, muscle blood restriction during an incremental cycling exercise test significantly limited exercise performance. Exercise intolerance with muscle blood flow restriction was mainly associated with attenuated cardiac responses despite cerebral activation reaching similar maximal levels as without muscle blood flow restriction.’
Reviewer’s comment
Unfortunately, I did not understand well about results of the “enhanced cerebral activation” as mentioned in line 43. In my knowledge, authors said in previous draft like “unaltered central nervous system…”. Please make sure of your results and interpretation.
Response:
We thank the reviewer for their comment. We would like to mention that cerebral activation as assessed by EEG activity was statistically similar between experimental conditions without and with muscle blood flow restriction despite the marked differences in exercise time (~ 28%). Based on these findings, it suggests that at exhaustion cerebral activation with muscle blood flow restriction was maximal and identical to that attained during exercise without muscle blood flow restriction and achieved in a shorter time period. This means, from a physiological point of view, that muscle blood flow restriction accelerates the rate of increase in cerebral activation. Furthermore, we have modified the ‘enhanced cerebral activation’ to ‘maximal cerebral activation’ or ‘earlier reach of peak’ or/and ‘accelerate the rate of increase in cerebral activation’ throughout the revised manuscript.
Minor
Reviewer’s comment
In the present study, authors found that muscle local occlusion resulted in lower muscle oxygenation level, increased blood pressure, increased perceived exertion level, increased cerebral activation level (?), and lower cardiovascular responses as described in simple summary. However, in line 21, authors deleted cardiovascular responses as important determinants, but again appeared in line 43 in abstract. Is it important result or not? Does cardiovascular system include blood pressure or not? Does “central nervous system” (line 21) include cerebral activation (line 43)? How about HR? is this a part of central nervous system (under autonomic nerve)? Please categorize properly. It is very difficult to follow authors’ intention.
Response:
We thank the reviewer for their comment. As we mentioned before, muscle blood flow restriction accelerates the rate of increase in cerebral activation and thus the central nervous system activation. Furthermore, we would like to mention that attenuated cardiovascular response with muscle blood flow restriction is a possible determinant of maximal aerobic capacity and we have mentioned that both in simple summary and abstract. In the present study, we found that muscle blood flow restriction accelerates the rate of increase in blood pressure response and cerebral activation, and as a consequence at exhaustion, blood pressure response and cerebral activation was maximal and identical in both condition despite marked differences in exercise time and at the same time the cardiovascular response was submaximal. We emphasized this phenomenon that when these parameters reach a predetermined maximal level (blood pressure response and cerebral activation), the exercise ability might be limited to maintain body’s integrity which in turn could modify the cardiac response, and then exercise is terminated. In addition, the increased cerebral activation means that central nervous system activation was also increased. Cerebral activation incorporates EEG activity, RPE, and EMG activity (we have mentioned it throughout the revised manuscript). Furthermore, the heart rate is a component of both the autonomic nervous system and the peripheral cardiovascular system, and it cannot be assessed without knowledge of the autonomic nervous system's response. In addition, we would like to mention that when we investigate the heart rate, stroke volume, and cardiac output during exercise we talk about cardiac responses, whereas when investigating the blood pressure response we talk about cardiovascular responses. We have modified the aforementioned terms throughout the revised manuscript.
Reviewer’s comment
Line 24. “This experimental approach could be used in the investigation of the mechanisms of cardiovascular dysfunction during exercise.”
Reviewer think this sentence seems difficult to be understood by journal readers if readers read only “simple summary”. What does “dysfunction” mean? exhausted condition in exercise as normal regulation?? If authors want to try picking up the usefulness of your model, apparently explanation is deficient in simple summary. Please again re-consider your point to be emphasized here.
Response:
Following the reviewer's suggestions, we have reconsidered the last sentences in the simple summary of the revised manuscript (page 1, lines 23-25) which now reads as follows ‘Muscle blood flow restriction might simulate pathological conditions such as heart failure, hypertension, and peripheral vascular diseases, which are all characterized by skeletal muscle perfusion impairments and, thus provide insights into exercise intolerance in health and disease.’
Reviewer’s comment
In line 59, authors divided factors into central vs. peripheral. Please make sure again whether a) to d) factor is “central” factor or not.
Response:
We thank the reviewer for their comment. We would like to mention that according to the bibliography, VO2max limiting factors have been classified into two main categories: central and peripheral. The central factors mainly referred to the oxygen delivery system, while the peripheral factors referred to the oxygen consumption system. Therefore, following the above classification, the factors a) oxygen carrying capacity of blood, b) pulmonary diffusion capacity, c) maximal cardiac output, and d) muscle blood flow are central factors. In addition, the term ‘central’ also refers to the possible contribution of the central nervous system or cerebral activation in the limitation of the exercise capacity.
We have reconsidered this paragraph in the revised manuscript to avoid confusion with the terms which now reads as follows (page 3, lines 58-70) ‘The V ̇O2max limiting factors have been consistently studied since 1924 but there is still controversy over which factor is the critical one. Exercise performance during incremental exercise to exhaustion relies on multiple and diverse facets which have been classified into two main categories: oxygen delivery system and oxygen consumption system. Factors described to be associated with oxygen delivery system limitation include the a) oxygen carrying capacity of blood [9], b) pulmonary diffusion capacity [10], c) maximal cardiac output [11,12,13], d) muscle blood flow [14,15], and e) cerebral blood flow and oxygenation [16,17,18], while oxygen consumption limitation refers to a) muscle oxygen diffusing capacity [19], b) mitochondrial enzymes levels [20], and c) muscle capillary density [21]. Furthermore, central command has been shown to contribute to the limitation of exercise capacity through modification of cerebral activation, and rate of perceived exertion [22]. It is worth noting that exaggerated muscle afferents group III and IV feedback to the central nervous system has been also implied as a limiting factor [23,24,25].’
Reviewer’s comment
In line 90, “paradoxically”?
Response:
We thank the reviewer for their comment. We have modified the ‘paradoxically’ to ‘however’ in the revised manuscript (page 3, line 95) to emphasize the fact that although the VO2max limiting factors have been consistently studied since 1924, there is only one study so far investigating the effects of restricted skeletal muscle oxygenation on maximal aerobic capacity.
Reviewer’s comment
Ref 36 and 37 are published by your group. Please clearly mention it in the text.
Response:
We thank the reviewer for their comment. Following the reviewer’s suggestion, we have specified in the revised manuscript (page 3, line 95, page 4, line 114) that references 36 and 37 in the revised manuscript (page, line) refer to investigations conducted by our laboratory.
Reviewer’s comment
Line 125: “for the first time in a holistic and integrative approach” seems abstract expression. Please have re-consideration.
Response:
We thank the reviewer for their comment. Following the reviewer’s suggestion, we have reconsidered the above sentences in the revised manuscript (page 4, line 127) which now reads as follows ‘in a novel and comprehensive manner’.
Reviewer’s comment
Line 148: “Participants visited the laboratory three times to become familiar with the experimental procedures” means total 3 visiting (practice, task1, task2) or total 5 visiting (3-times practice, task1, task2)?
Response:
We thank the reviewer for their comment. We would like to mention that the participants visited the laboratory a total of 3 times, once for familiarization and two for the experimental conditions without and with muscle blood flow restriction. We have provided the above information in the revised manuscript (pages 4-5, lines 151-157).
Reviewer’s comment
Line 216: “Muscle and cerebral measurements were normalized to zero before recording baseline values to express the magnitude of muscle and cerebral deoxygenation at rest and maximal exercise without and with muscle blood flow restriction.”
What does “baseline” mean? When did you measure it? before occlusion or after occlusion? Please mention it clearly.
If “after occlusion”, why did you not use the before-occlusion value as zero point. Please give me your comment.
Response:
We thank the reviewer for their comment. We would like to mention that baseline values were recorded for five minutes after the stabilization of the hemodynamic responses, while the participant was resting in a back-supporting chair with the legs bent at the hip and knee (~90o) as we mentioned in the revised manuscript (page 5, lines 166-168). Furthermore, the above information is demonstrated clearly in the new Figure 1 of the overview of the experimental protocol of the current study (see Figure 1). Thus, the baseline values are recorded at rest without muscle blood flow restriction and before thigh cuffs application.
Reviewer’s comment
In line 329, authors mentioned “Specifically, muscle Δ[O2Hb] gradually decreased from the beginning of exercise to 90%PPO and then remained low (Figure 1, A1, B1)”. However, X axis of A1 is not %PPO. Also, at the 100%, I can see further reduction after 90%. Please make sure.
Response:
We thank the reviewer for their comment. Indeed, the Figure 1 B1 referred to %PPO, whereas the Figure 1 A1 referred to absolute workload and we have corrected this in the revised manuscript (page 9, line 340). We would like to mention that muscle oxygenation results are now demonstrated in Figure 2 in the revised manuscript as we added a new Figure 1 with the schematic design of the study. Furthermore, we have removed the muscle oxygenation response during incremental exercise at isotime-isowatt (A) and we kept only in the relative work rate In addition, according to the statistical analysis, there is no significant difference between 90% and 100%PPO for muscle Δ[O2Hb] (p=0.365).
Reviewer’s comment
Line 348: n=13, not n-13.
Response:
Following the reviewer’s suggestion, we have corrected the ‘n-13’ to ‘n=13’ in the revised manuscript (page 10, line 357).
Reviewer’s comment
Line 333: The magnitude of muscle Δ[O2Hb] decline was significantly greater in NC condition from 60-100%PPO (p<0.001), while the muscle Δ[THb] rise was significantly greater in the WC condition from 20-100%PPO (p<0.001).” Authors may be talking about fig B1 for the Δ[O2Hb], is the right? Which fig is for THb? By any chance, are you talking about Δ[HHb], not THb? Because I can see†symbol on 20-100%PPO in fig B2, not in B3. Please make sure and demonstrate clearly.
Response:
We thank the reviewer for their comment. We would like to mention that the magnitude of muscle Δ[O2Hb] changes demonstrated in the Figure 2 B1 (%PPO), whereas the muscle Δ[THb] changes demonstrated in the Figure 2 B3 (%PPO). In addition, the muscle Δ[HHb] changes demonstrated in the Figure 2 B2 (%PPO) (in the revised manuscript is in Figure 2 B) and not in Figure 2 B3. As we mentioned the significant † symbol on 20-100%PPO is illustrated in Figure 2 B3 (in the revised manuscript is in Figure 2 C). We have provided the above information clearly in the revised manuscript (page 9, lines 338-346).
Reviewer’s comment
Line 373: “Specifically, cerebral-Δ[HHb] was unchanged at the initiation of exercise and rapidly increased at 90%PPO and then stabilized until the end of exercise.”
90%?, not 80%? in fig 2B2.
Response:
We thank the reviewer for their comment. Indeed, cerebral Δ[HHb] rapidly increased at 80%PPO and not at 90%PPO and then stabilized until the end of the exercise. We apologize for our misleading and we have corrected it in the revised manuscript (page 11, line 379).
Reviewer’s comment
In the present study, authors used specific model of local occlusion to overcome experimental problem obtained in environmental (e.g., hypoxic) experiment. In line 80-88, authors have mentioned usefulness of experimental model. At now, after present experiment, how should we think advantage of local occlusion model? is it possible to apply present specific results to physiology appeared under regular (normal) exercise? What is the difference among exercise-induced fatigue, exercise- and hypoxia-induced fatigue, and occlusion-induced fatigue? Please give us your summary about experimental model itself. This may contribute to increase scientific value of present study.
Response:
We thank the reviewer for their comment. We would like to mention that in the current study, we had two separate conditions, without muscle blood flow restriction (control condition – normal exercise) and with muscle blood flow restriction (experimental condition). The comparative results of this study provide information about the limiting factors of exercise tolerance in both normal condition (without muscle blood flow restriction) and experimental condition (with muscle blood flow restriction). Essentially, muscle blood flow restriction accelerates the onset of fatigue by causing premature peak stimulation of the central nervous system. Since minimum muscle oxygenation indexes and maximum cerebral activation coincided at exercise cessation regardless of experimental condition, the conclusion that muscle oxygenation is the prime trigger of exercise termination, no matter what could be drawn. For the first time, however, it was shown that despite muscular activity and cerebral activation reaching their limits, cardiac never approaches peak performance. The results of the present experimental paradigm look alike to the results derived from the hypoxia-induced fatigue where hypoxemia even in the brain is present. This fact confirms that muscle oxygenation sets the rate of fatigue development. Therefore, the significance of the present study was to highlight the importance of skeletal muscle oxygenation as a possible determinant of exercise tolerance during maximal whole-body exercise, as it accelerates cerebral activation to reach its peak while impairing cardiac peak performance. Furthermore, our results provide considerable insight into the physiological responses during incremental exercise to exhaustion with muscle blood flow restriction in young athletes, in whom this intervention is often used to enhance exercise performance. Finally, exercise intolerance in conditions such as chronic heart failure, hypertension, peripheral vascular diseases, and vascular claudication, which are characterized by skeletal muscle perfusion impairments, compromised oxygenation, and blood pooling, can be considered through the simulation of muscle blood flow restriction in healthy individuals.
We have provided the above information in the revised manuscript (page 21, lines 728-739) which now reads as follows ‘Thus, our findings highlight the importance of skeletal muscle oxygenation as a possible determinant of exercise tolerance during maximal whole-body exercise, as it accelerates cerebral activation to reach its peak and impairs maximal cardiac responses. Further research is required to elucidate the causal link between cerebral and cardiac cross communication during blood flow restriction. Furthermore, our results provide considerable insight into the physiological responses during incremental exercise to exhaustion with muscle blood flow restriction in young athletes, in whom this intervention is often used to enhance exercise performance. Finally, exercise intolerance in conditions such as chronic heart failure, hypertension, peripheral vascular diseases, and vascular claudication, which are characterized by skeletal muscle perfusion impairments, compromised oxygenation, and blood pooling, can be considered through the simulation of muscle blood flow restriction in healthy individuals.’
Reviewer’s comment
In line 42, authors only mentioned “association” as parameters moving together under local occlusion. In contrast, line 21, it seems difficult to say whether these changes are important (critical) or not because authors just see movement of parameters. Please tune the tone of your statement/expression.
Response:
Following the reviewer’s suggestion, we have modified these lines in the simple summary of the revised manuscript (page 1, lines 19-21) which now reads as follows ‘These findings suggest that skeletal muscle oxygenation levels in combination with augmented blood pressure response and cerebral activation could be important determinants setting the limits of exercise performance.
Reviewer’s comment
Is there any relation between cerebral oxygenation level and cerebral EEG level? How does EEG move under environmental (e.g., hypoxia) challenge? Is this different from your experimental model? Please add information in the text.
Response:
We thank the reviewer for their comment. We would like to mention that in the current study, we did not find any significant correlation between cerebral oxygenation indices and EEG activity (p-values=0.08-0.974). However, cerebral oxygenation level has been shown that independently influence cerebral electrocortical activity and impaired exercise performance (Periad et al 2018, Acta Physiol (Oxf), 222(1). doi: 10.1111/apha.12916. Epub 2017 Jul 25; Goodall et al 2014, Fatigue, 2(2):73-92. doi: 10.1080/21641846.2014.909963; Rasmussen et al 2010, J Physiol, 588(Pt 11):1985-95. doi: 10.1113/jphysiol.2009.186767. Epub 2010 Apr 19.). We would like to mention that in the aforementioned studies were conducted during acute or/and chronic arterial systemic hypoxia which is different from local hypoxia where systemic oxygen availability is maintained.
Reviewer’s comment
In line 106, suddenly “cerebral activation” and “oxygenation” came up to introduction as factors for fatigue/vo2max-limiting factor. How does it affect on fatigue? Please add additional brief explanation at first appearance.
Response:
We thank the reviewer for their comment. As previously mentioned, we have rephrased the VO2max limiting factors in the revised manuscript (page 3, lines 58-70) and referred cerebral activation and cerebral oxygenation as possible determinants of exercise performance.
Furthermore, inhibition of cerebral activation and cerebral oxygenation impairments have been demonstrated as a limiting factor of exercise performance during incremental exercise to exhaustion due to inadequate motor unit recruitment causing premature fatigue [Rasmussen et al 2020; J Physiol, 588(11), 1985-1995. doi: 10.1113/jphysiol.2009.186767.77].
Reviewer’s comment
In figures, literature size seems very small. Would you kindly re-arrange figures?
Response:
We thank the reviewer for their comment. Following the reviewer’s suggestion, we have re-arranged the figures in the revised manuscript (please see the Figures 1-6). We would like to mention that the biology template is so strict that the figures are automatically shrinking so we additional uploaded the original figures to adjust properly.
Reviewer’s comment
In the present study, results are expressed in relative and absolute manner, at some sampling point (baseline, rest, during exercise, exhaustion) with comparison to apposite group, previous sampling point. So, it is difficult to follow and understand results. Especially, concept/interpretation against during exercise and at exhaustion, and against relative and absolute expression seems have some difficulty in adequate understanding. Reviewer requests additional arrangement/ ingenuity to your presentation.
Response:
We thank the reviewer for their comments. We would like to mention that following the reviewer’s suggestion, we have tried to improve the manuscript.

Reviewer 2 Report (New Reviewer)
Dear Authors,
Manuscript ID: biology- 2447667
Title Manuscript: The effect of skeletal muscle oxygenation on hemodynamics, cerebral oxygenation and activation, and exercise performance during incremental exercise to exhaustion in healthy individuals
This is an important topic since the study participants are cyclists and triathletes with mean VO2max > 50 ml/kg/min but at the moment MAJOR REVISIONS are necessary in order to make it suitable for a final decision for “Biology”;
This study examined the effect of skeletal muscle oxygenation on hemodynamics, cerebral oxygenation and activation, and exercise performance during maximal incremental exercise test to exhaustion on a bicycle ergometer in two experimental conditions (muscle blood flow restriction and without restriction) in male cyclists and triathletes.
POINTs of STRENGTH:
1) The effects of muscle blood flow restriction and without restriction interventions on hemodynamics, cerebral oxygenation, cerebral activation, and exercise performance during maximal incremental exercise test in male cyclists and triathletes;
POINTs of WEAKNESS (and/or should be revised to improve the manuscript):
Main Title
2) Please modify the main title of this study as follows:
The effect of skeletal muscle oxygenation on hemodynamics, cerebral oxygenation and activation, and exercise performance during incremental exercise to exhaustion in male cyclists and triathletes
Abstract
3) Please add gender, mean weight and BMI for participants in the “methods” section of the abstract;
4) The significance level of results is unclear. Please clarify in the results section;
Keywords
5) Please modify the keywords section as follows:
Incremental exercise test; Muscle blood flow restriction; Muscle oxygenation; Cerebral oxygenation; Cardiorespiratory fitness; Cyclists
1. Introduction
6) Please modify the “healthy individuals” [line 128] to “male cyclists and triathletes” at the end of the introduction section of this study;
2. Materials and Methods
2.1. Participants
7) The recruitment process and/or screening of study participants, especially inclusion and exclusion criteria should be described in more detail such as initial sample size, gender, age range, BMI category based on the WHO, blood pressure, free of medications and/ or drug intervention, physical fitness level, healthy status, and so on.
2.3. Experimental Protocol
8) Please provide a schematic design for the experimental protocol of this study;
2.4.6. Data Analysis and Statistics
9) Did authors use a statistical software to calculate the sample size? If YES, please explain and add its name and valid reference in the “data analysis and statistics” section;
10) The significance level of statistical analysis considered for one-tailed OR two-tailed? Please clarify;
3. Results
11) Results section is well-written;
4. Discussion and Conclusion
12) Please modify the “healthy individuals” to “male cyclists and triathletes” in line-500 of the discussion section as well as in line-742 of conclusion section;
13) As mentioned above, the authors will agree that the limitations section has to be expanded;
14) What does this study add to the literature? Please explain and add in the conclusions section;
15) Please provide applied approaches of blood flow restriction for athletes such as cyclists and triathletes at the end of this manuscript;
References
16) References are not always in accordance with the authors' guidelines. In particular, please check No. 29, 30, 31, 38, 57, and 94 for validation.
Best Regards
15 June 2023
Author Response
REVIEWER'S COMMENTS to Author:
Reviewer: 2
Reviewer’s comment
This is an important topic since the study participants are cyclists and triathletes with mean VO2max > 50 ml/kg/min but at the moment MAJOR REVISIONS are necessary in order to make it suitable for a final decision for “Biology”;
This study examined the effect of skeletal muscle oxygenation on hemodynamics, cerebral oxygenation and activation, and exercise performance during maximal incremental exercise test to exhaustion on a bicycle ergometer in two experimental conditions (muscle blood flow restriction and without restriction) in male cyclists and triathletes.
Reviewer’s comment
POINTs of STRENGTH:
1) The effects of muscle blood flow restriction and without restriction interventions on hemodynamics, cerebral oxygenation, cerebral activation, and exercise performance during maximal incremental exercise test in male cyclists and triathletes;
Response:
We thank the reviewer for his/her positive comments on our work.
Reviewer’s comment
POINTs of WEAKNESS (and/or should be revised to improve the manuscript):
Main Title
2) Please modify the main title of this study as follows:
The effect of skeletal muscle oxygenation on hemodynamics, cerebral oxygenation and activation, and exercise performance during incremental exercise to exhaustion in male cyclists and triathletes
Response:
We would like to mention that the majority of the participants were cyclists and not triathletes so we have followed the reviewer’s suggestion and we have modified the main title of this study which now reads as follows ‘The effect of skeletal muscle oxygenation on hemodynamics, cerebral oxygenation and activation, and exercise performance during incremental exercise to exhaustion in male cyclists’
Reviewer’s comment
Abstract
3) Please add gender, mean weight and BMI for participants in the “methods” section of the abstract;
4) The significance level of results is unclear. Please clarify in the results section;
Response:
We thank the reviewer for their comment. Following the reviewer’s suggestion, we have inserted the above information in the revised manuscript (page 1, lines 28-30) which now reads as follows ‘Thirteen healthy males, cyclists (age 33±2 yrs., body mass: 78.6±2.5 kg, and body mass index: 25.57±0.91 kg·m-1), performed a maximal incremental exercise test on a bicycle ergometer in two experimental conditions.’ Furthermore, we have clarified the significance level of the results in the revised manuscript (page 1, lines 32-42).
Reviewer’s comment
Keywords
5) Please modify the keywords section as follows:
Incremental exercise test; Muscle blood flow restriction; Muscle oxygenation; Cerebral oxygenation; Cardiorespiratory fitness; Cyclists
Response:
Following the reviewer’s suggestion, we have modified the keywords sections in the revised manuscript which now reads as follows ‘Incremental exercise test; Muscle blood flow restriction; Muscle oxygenation; Cerebral oxygenation; Cardiorespiratory fitness; Cyclists’.
Reviewer’s comment
- Introduction
6) Please modify the “healthy individuals” [line 128] to “male cyclists and triathletes” at the end of the introduction section of this study;
Response:
Following the reviewer’s suggestion, we have modified the ‘healthy individuals’ to ‘male cyclists’ in the revised manuscript (page 4, line 125) as the majority of the participants were cyclists and not triathletes.
Reviewer’s comment
- Materials and Methods
2.1. Participants
7) The recruitment process and/or screening of study participants, especially inclusion and exclusion criteria should be described in more detail such as initial sample size, gender, age range, BMI category based on the WHO, blood pressure, free of medications and/ or drug intervention, physical fitness level, healthy status, and so on.
Response:
We thank the reviewer for their comment. Following the reviewer’s suggestion, we have described in more detail the inclusion criteria and the demographic characteristics of the participants in the revised manuscript (page 4, lines 132-141) which now reads as follows ‘Inclusion criteria were as follows: (1) only male participants, (2) recreational cyclists or triathletes with at least five years of specific cycling training, (3) nonsmokers, (4) no restraining cardiovascular, respiratory, musculoskeletal or neurological diseases, (5) age above 18 yrs, (6) no drug abuse or use of medications known to affect exercise performance, and (7) ability to follow the procedures of the current study and visit our laboratory totally 3 times. All participants (n=13) who met the inclusion criteria voluntarily participated in this study and completed the experimental procedures. Their mean (±SEM) age, body mass, stature height, body fat, and O2max were 33±2 yrs. (18-45 yrs.), 78.6±2.5 kg (63.6-93.7 kg), 176.0±1.9 cm (163-188 cm), 12.4±1.5 % (4.45-17.69%), and 50.5±2.2 ml∙kg-1∙min-1 (39.24-65.17 ml∙kg-1∙min-1), respectively.’
Reviewer’s comment
2.3. Experimental Protocol
8) Please provide a schematic design for the experimental protocol of this study;
Response:
We thank the reviewer for their comment. Following the reviewer’s suggestion, we have provided a schematic design of the experimental protocol of the current study in the revised manuscript which is the following:
Figure 1. Overview of the experimental protocol which consisted of an incremental exercise test to exhaustion on a cycle ergometer on two separate conditions: a) without (No Cuffs, NC) and b) with (With Cuffs, WC) muscle blood flow restriction. Brain electroencephalography activity (EEG), skeletal muscle activity (EMG), muscle oxygenation, cerebral oxygenation, cardiopulmonary, and perceptual responses were measured during the exercise protocol. Maximal voluntary isometric contraction (MVC) of the knee extensors was evaluated before and immediately after the end (~ 2.5 min) of the incremental exercise test. PPO: peak power output.
2.4.6. Data Analysis and Statistics
9) Did authors use a statistical software to calculate the sample size? If YES, please explain and add its name and valid reference in the “data analysis and statistics” section;
Response:
We thank the reviewer for their comment. Unfortunately, we would like to mention that we did not use any statistical software to calculate the sample size of the present study.
Reviewer’s comment
10) The significance level of statistical analysis considered for one-tailed OR two-tailed? Please clarify;
Response:
We thank the reviewer for their comment. We would like to mention that the significance level of statistical analysis was considered for two-tailed. We have provided this information in the revised manuscript (page 8, lines 318-319) which now reads as follows ‘. Values are reported as mean ± standard error of measurements (SEM) and the two-tailed values of p<0.05 were considered statistically significant.’
Reviewer’s comment
- Results
11) Results section is well-written;
Response:
We thank the reviewer for his/her positive comment.
Reviewer’s comment
- Discussion and Conclusion
12) Please modify the “healthy individuals” to “male cyclists and triathletes” in line-500 of the discussion section as well as in line-742 of conclusion section;
Response:
Following the reviewer’s suggestion, we have modified the ‘healthy individuals’ to ‘male cyclists’ in the revised manuscript (page 15, line 476, and page 19, line 725).
Reviewer’s comment
13) As mentioned above, the authors will agree that the limitations section has to be expanded;
Response:
We thank the reviewer for their comment. Following the reviewer’s suggestion, we have expanded the limitations section in the revised manuscript (page 19, lines 704-713) which now reads as follows ‘Furthermore, cuff occlusion pressure was set at 120 mmHg for all participants and did not individualize according to the individual arterial occlusion pressure. The absolute cuff occlusion pressure value of 120 mmHg has been shown to elicit approximately 55-65% blood flow restriction [50]. These blood flow reductions are widely used in resistance and aerobic exercise intervention with muscle blood flow restriction to enhance skeletal muscle and aerobic adaptations in health and diseases [29,30,31,32]. Other limitations of this study are that it included only young healthy male athletes with high physical fitness. Thus, these results may not necessarily apply to individuals with different characteristics. Future studies should include nonathletes, females, and patients with chronic diseases.’
Reviewer’s comment
14) What does this study add to the literature? Please explain and add in the conclusions section;
Response:
We thank the reviewer for their comment. We would like to mention that our findings highlight the importance of skeletal muscle oxygenation as a possible determinant of exercise tolerance during maximal whole-body exercise, as it drives cerebral activation to peak value earlier and impairs maximal cardiac responses. Furthermore, our results provide considerable insight into the physiological responses during incremental exercise to exhaustion with muscle blood flow restriction in young athletes, in whom this intervention is often used to enhance exercise performance. Finally, muscle blood flow restriction in healthy individuals simulates pathological conditions such as chronic heart failure, hypertension, peripheral vascular diseases, and vascular claudication, which are characterized, as in the present experimental paradigm, by skeletal muscle perfusion impairments, tissue hypoxemia, and venous blood pooling, and provides insight into physiological parameters which should be treated in order to alleviate such adverse conditions. We have provided the above information in the revised manuscript (pages 19-20, lines 728-739) which now reads as follows ‘Thus, our findings highlight the importance of skeletal muscle oxygenation as a possible determinant of exercise tolerance during maximal whole-body exercise, as it accelerates cerebral activation to reach its peak and impairs maximal cardiac responses. Further research is required to elucidate the causal link between cerebral and cardiac cross communication during blood flow restriction. Furthermore, our results provide considerable insight into the physiological responses during incremental exercise to exhaustion with muscle blood flow restriction in young athletes, in whom this intervention is often used to enhance exercise performance. Finally, exercise intolerance in conditions such as chronic heart failure, hypertension, peripheral vascular diseases, and vascular claudication, which are characterized by skeletal muscle perfusion impairments, compromised oxygenation, and blood pooling, can be considered through the simulation of muscle blood flow restriction in healthy individuals.’
Reviewer’s comment
15) Please provide applied approaches of blood flow restriction for athletes such as cyclists and triathletes at the end of this manuscript;
Response:
We would like to mention that the aim of this study was not to provide practical consideration of muscle blood flow restrictions for athletes. However, following the reviewer’s suggestion, we have provided some information in the revised manuscript (page 20, lines 740-743) which now reads as follows ‘In recent years, athletes of various fitness levels are currently using this method (muscle blood flow restriction) at rest, known as ischemic preconditioning, and during resistance and aerobic exercise training protocols as an alternative intervention to improve exercise performance and enhance skeletal muscle and aerobic adaptations.’
Reviewer’s comment
References
16) References are not always in accordance with the authors' guidelines. In particular, please check No. 29, 30, 31, 38, 57, and 94 for validation.
Response:
We thank the reviewer for their comment. We have checked and revised the aforementioned references according to the authors’ guidelines.

Round 2
Reviewer 1 Report (Previous Reviewer 1)
Thank you for your considerable effort for revise. Authors have responded to all of reviewer’s requests. I am sure that manuscript has reached the publication level. Thank you for your cooperation.
Reviewer 2 Report (New Reviewer)
Dear Authors,
Manuscript ID: biology- 2447667
Title Manuscript: The effect of skeletal muscle oxygenation on hemodynamics, cerebral oxygenation and activation, and exercise performance during incremental exercise to exhaustion in male cyclists
I am very grateful to the authors for their efforts.
In general, this manuscript has found suitable content after correcting major revisions, and the modified revisions are accepted.
Best Regards
29 June 2023
This manuscript is a resubmission of an earlier submission. The following is a list of the peer review reports and author responses from that submission.
REVIEWER'S COMMENTS to Author:
Reviewer: 1
Reviewer’s comment
Comments and Suggestions for Authors
To authors:
This study examined the effect of local muscle occlusion on exercise performance with simultaneous measurement of oxygen-associated parameters in brain, heart, and skeletal muscle. And authors concluded that the occlusion resulted in a lower exercise performance, and this might be due to reduced cardiovascular response.
Experimental methods itself are simple, and a large part of methods are standard techniques, so results provided seems reliable. In contrast, basically, outstanding results are not much nevertheless of considerable results. Authors had better to emphasis your priority by using sentences like “This is first evidence…” or “We showed at the first time…”.
Response:
We thank the reviewer for his/her comments.
Following the reviewer’s suggestion, we have highlighted the main findings of the study and indicated in the revised manuscript that these are originals and prototypes.
Please see comments below.
Major:
Reviewer’s comment
Please make sure what is your real purpose and conclusion. For example, in line 20, your target was to examine the effect of occlusion on exercise performance but in line 33, conclusion said what is associated event with occlusion. The conclusion seems not to answer to the purpose properly. Also, in line 107, authors again mentioned that one of purpose is to see whether occlusion effect on performance but as authors mentioned it was already reported in ref 47. As same with this, authors can expect that occlusion would blunt cardiovascular response as shown in ref 47(this is not new), but I think in this experimental condition, it seems impossible to identify whether the blunt cardiovascular response would be the definitive cause or not as in line 112 nevertheless of results. In think adequate “tone” of your real purpose might be only to add information about brain oxygenation and to see event happen parallelly. Overall, purpose/importance seems very weak. Authors have to pay more attention to highlight your originality and its importance with better presentation.
Response:
We thank the reviewer for his/her comments.
We agree with the reviewer’s concerns that the purpose of this study was not clearly defined and there is a lack of consistency in the conclusion statement both in the abstract and main text. In the revised manuscript we have accurately stated the purpose of this study and the conclusion is consistent with the purpose [please see simple summary (page 1, lines 13-26), abstract (page 1, lines 27-44), introduction (page 3, lines 125-131), and discussion section (page 15, lines 498-518) in the revised manuscript].
Concerning the effect of thigh cuffs application on exercise performance, we would like to mention that although muscle blood flow restriction during exercise, either resistance or aerobic, is thoroughly applied in exercise programs to promote muscle and aerobic adaptations in health and diseases, only one study (Geladas et al 2009) has been conducted so far investigating the effects of restricted skeletal muscle oxygenation through thigh cuffs applications on maximal aerobic capacity. However, in the aforementioned study, there are several limitations. Specifically, the sample size was only 6 young male participants with an average exercise capacity (VO2max: 42.9±3.7 ml/kg/min). Therefore, we first, aimed to confirm the results of the adverse effect of skeletal muscle blood flow and oxygenation reduction on exercise performance as evaluated by the incremental exercise test to exhaustion and then to investigate the underlying factors that may explain the development of exercise intolerance. Furthermore, in the study of Geladas et al (2009), it was not possible to measure the entire cardiovascular response to incremental exercise to exhaustion except for the measurement of heart rate. It is well established that the oxygen delivery system is one of the determinant factors of aerobic capacity. Thus, the recording of the cardiac output, stroke volume, peripheral resistance, and perfusion pressure (systolic, diastolic, and mean arterial pressure) throughout exercise are really important to explain exercise intolerance. Besides this, blood pressure response to exercise monitoring is an essential consideration as skeletal muscle oxygenation and blood flow reduction, and venous occlusion has been demonstrated to exaggerate blood pressure response associated with peripheral vasoconstriction due to activation of group III/IV muscle afferents, a factor that may explain the impairment of exercise tolerance with muscle blood flow restriction. In addition, cerebral activation fluctuations such as the rate of the motor unit recruitment, cerebral electrocortical activity, and cerebral oxygenation were not taken into consideration during incremental exercise with muscle blood flow restriction except for the perceptual response in the study of Gelada et al (2009). Thus, for a comprehensive overview of the limiting factors of exercise performance during exercise with muscle blood flow restriction, this study was conducted to investigate all possible factors that could explain the lower maximal oxygen consumption.
Additionally, we have provided the above information in the revised manuscript (pages 3-4, lines 80-113) which now reads as follows ‘In recent years, muscle blood flow restriction during resistance and aerobic exercise has been thoroughly applied in exercise training protocols as a novel intervention to enhance skeletal muscle and aerobic adaptations in healthy individuals, as well as in patients with chronic disease [29,30,31,32]. This intervention relies on the application of external pressure over the proximal limb musculature (upper and lower), with the intent to reduce arterial inflow, restrict venous blood flow, cause blood pooling in capacitance vessels, and accumulate the metabolic by-products within the exercising skeletal muscles [33,34,35]. Thus, muscle blood flow restriction during exercise is thought to induce a reduction in skeletal muscle oxygenation (tissue hypoxia), while systemic oxygen availability is maintained.
Paradoxically, only one study [36] has been conducted so far investigating the effects of restricted skeletal muscle oxygenation through thigh cuffs application on maximal aerobic capacity and found that exercise performance, as defined by O2max and peak power output (PPO) in incremental exercise, were significantly reduced. Furthermore, these results were accompanied by a similar level of skeletal muscle deoxygenation and perceptual exertion at task failure despite the marked differences in exercise tolerance, whereas heart rate was significantly lower in the muscle blood flow restriction condition. Thus, the impairment in exercise tolerance with muscle blood flow restriction was only associated with a maximal heart rate reduction suggesting cardiac limitations. However, it was not possible to untangle whether the lower maximal heart rate (HRmax) was due to the lower PPO produced or vice versa. This is a crucial question because there is a long-standing debate about whether aerobic capacity is limited by the cardiovascular or central nervous system (REF?). Furthermore, in the aforementioned study [36], there are several limitations. Specifically, the sample size was only six young male participants with an average exercise capacity ( O2max: 42.9±3.7 ml/kg/min). In addition, the cardiovascular responses to incremental exercise to exhaustion were not thoroughly investigated except for HR recording. In addition, cerebral activation and oxygenation were not taken into consideration during incremental exercise with muscle blood flow restriction except for the perceptual response of fatigue. Recently, Cherouveim et al. (2021) [37] showed that muscle oxygenation reduction through venous occlusion even at rest can be sensed by the central nervous system (CNS) probably via muscle afferents group III/IV feedback, eliciting changes in cerebral oxygenation/activation. However, to our knowledge, the effect of muscle blood flow restriction on cerebral activation and oxygenation during whole-body dynamic exercise has not been investigated yet.
Furthermore, following the reviewer’s suggestion we have highlighted the cerebral responses to muscle blood flow restriction during incremental exercise to exertion as to our knowledge no studies evaluated the cerebral oxygenation and cerebral activation during whole-body exercise with muscle blood flow restriction (please see the introduction; page 4, lines 106-113 and discussion section; pages 16-17, lines 582-613) in the revised manuscript). However, it is worth mentioning that the purpose of this study was to evaluate whether skeletal muscle oxygenation reduction via thigh cuffs application would affect the hemodynamic, cerebral oxygenation, and activation during incremental exercise to exertion in healthy individuals and whether the above changes would affect the whole-body exercise performance and we have to discuss it in some details.
Reviewer’s comment
Reviewer can agree that local occlusion model has some advantages to reveal mechanisms of physiological events. But unfortunately, as same as other model like hypoxia, some points were different from normal exercise condition. For example, occlusion could restrict returning blood flow volume to heart and this could affect on cardiovascular regulation. In contrast to exercise, under occlusion endocrinal/autonomic regulation of vascular tone does not have much meaning because physically blood flow is restricted by occlusion. Also, this stagnated liquid flow might facilitate changes of osmotic pressure, which is known as regulator of muscle metabolites including glycogen, ATP, PCr in an oxygen-independent manner. Furthermore, occlusion can affect endocrine system including growth hormone release, which is known as metabolic regulator. In line 101, authors mentioned “relative importance” among brain, heart, skeletal muscle but I think still it is not clear the relative importance and whether your results obtained in occlusion is capable to replace to normal exercise-associated events. Authors have to mention your experimental limitation with caution for avoiding overestimation.
Response:
We accept the reviewer’s comment and acknowledge the limitation of the results of the used model. Please see (page 19, lines 720-726) which now reads as following ‘A special methodological issue is that venous occlusion and blood pooling to the lower extremities affects a wide range of physiological responses consequently producing different exercise conditions from those without muscle blood flow restriction. Thus, these results may not necessarily apply under physiological exercise conditions. However, we used the muscle blood flow restriction intervention during exercise to restrict skeletal muscle oxygenation and thus to stress the O2 delivery system to investigate the possible determinants of exercise tolerance.’
Furthermore, we agree with the reviewer that the relative importance of cerebral activation, cardiovascular response, and skeletal muscle on exercise performance during incremental exercise to exhaustion is still unclear. In the revised manuscript, we have modified the above statement. However, as the reviewer states in his comments ‘that local occlusion model has some advantages to reveal mechanisms of physiological events’ this point was further clarified in lines 726-730 which now reads as following ‘It is worth mentioning that the muscle blood flow restriction method appears to induce physiological responses similar to those observed in chronic diseases such as hypertension, heart failure, and peripheral vascular disease [23,40,98]. Thus, muscle blood flow restriction intervention could be used in the future to investigate the development of cardiovascular diseases.’
Reviewer’s comment
In line 471, authors mentioned that skeletal muscle oxygenation is a crucial determinant in exercise performance, but in line 722, also mentioned that cardiovascular response is critical determinant. I could understand your point, but your presentation seems to be confusing for journal readers, especially abstract-readers. Are these parallel events or time-series events? Why authors think so? On the whole, presentation seems somewhat difficult to catch up points. Reviewer recommends that authors had better re-arrangement to make a logic in accordance with your purpose.
Minor:
Response:
Some events are serial, and some others are parallel. Muscular perturbations preceded cerebral events. Cardiac response to muscular perturbations and muscular performance should be parallel events. No matter whether the events are serial or parallel some of them could be crucial. We understand however the comment of the reviewer and we try in the revised version to express ourselves in a clear, not confusing, way. Please see page 15, lines 500-518 which now reads as follows ‘Furthermore, we elucidated whether the impairment of exercise performance with muscle blood flow restriction was associated with failure of central nervous system activation or/and inability of cardiovascular response to reach its maximum. Confirming our hypothesis, it was found that skeletal muscle oxygenation restriction via thigh cuffs application affected maximal aerobic capacity as indicated by a decrease in O2max, peak power output, and time to exhaustion. Furthermore, muscle blood flow restriction during whole-body dynamic exercise accelerated a) the rate of skeletal muscle deoxygenation, b) the rise in systolic blood pressure, rating of perceived exertion, and cerebral activation without these variables being different at the point of exhaustion, even though O2max, peak power output, and time to exhaustion were significantly impaired in WC compared to NC condition. These original data suggest that skeletal muscle oxygenation levels in combination with exaggerated blood pressure response and augmented central nervous system activation could be important determinants of exercise performance. Specifically, when the aforementioned parameters reach a maximal level, regardless of experimental condition, exercise effort terminates. Paradoxically, whereas arterial blood pressure reached its limit at the point of exercise cessation, the peak values of cerebral deoxygenation, regional cerebral blood volume, cardiac output, heart rate, and stroke volume were significantly compromised at task failure in muscle blood flow restriction compared with the control condition.
We thought that the rate of development of skeletal muscle deoxygenation could be the initial stimulus that triggers the exercise cessation via modification of either cerebral activation (central nervous system) or/and cardiovascular response. Indeed, at the same time, venous occlusion and blood pooling to the lower extremities enhanced cerebral activation and affected maximal cardiovascular responses (SVmax, HRmax, and Qmax). Thus, exercise capacity with muscle blood flow restriction is limited by the cardiac function and exhaustion is exhibited after insufficient oxygen delivery to the active skeletal muscle.
Reviewer’s comment
In Table and Figures, please re-consider adding rest value (after occlusion before exercise) and original value (before occlusion).
Response:
We agree with the reviewer’s concerns that resting values may be important to insert in the Table and Figures, but we believed that this would complicate the revised manuscript. In addition, the effect of muscle blood flow restriction on physiological responses at rest has been described in the results section. For a more detailed examination of the effects of blood flow restriction on physiological responses at rest, you may refer to the earlier publication by the same laboratory (Cherouveim et al 2021; Appl Physiol Nutr Metab. 46(10), 1216-1225. doi: 10.1139/apnm-2020-1082).
Reviewer’s comment
In line 12, is to standardize “model” your real purpose? If so, please put it in Abstract because in web PubMed, “Simple summary” does not come up. But again, please make sure whether method of “occlusion” is “new” or not. In comparison to the status before this experiment, did your results further standardize the model after your experiment? Please clear-cut your answer in the text in accordance with your purpose.
Response:
We would like to mention that the purpose of this study was to investigate the effect of reduced skeletal muscle oxygenation on hemodynamic responses, cerebral oxygenation, and cerebral activation during incremental exercise to exertion in healthy individuals. Furthermore, we elucidated whether the impairment of exercise performance with muscle blood flow restriction was associated with increased central nervous system activation or/and deterioration in cardiovascular response. That was the initial purpose and in the revised version this is clarified in a clear-cut way. We feel sorry for confusing the reader. This occurred because we misunderstood, the instructions to the authors' instructions, ‘the simple summary is not exactly either a short summary or short abstract of the study but a simple and concise description of the study to the public emphasizing how the conclusions from the study will be valuable to society’. We apologized for this misleading and have rewritten the simple summary in order to be more representative of the real purpose of this study in the revised manuscript (page 1, lines 13-26) which now reads as following: ‘This study aimed to elucidate whether exercise intolerance with muscle blood flow restriction was associated with increased central nervous system activation or/and deterioration in cardiovascular response. Muscle blood flow restriction (venous occlusion) during whole-body dynamic exercise accelerates a) the rate of skeletal muscle deoxygenation, b) the increase in systolic blood pressure, rating of perceived exertion, and cerebral activation without these variables to differ at exhaustion despite marked decrease in exercise time and maximal aerobic output was recorded. Maximal cardiovascular responses (i.e., heart rate, stroke volume, and cardiac output) were also significantly lower with muscle blood flow restriction. These findings suggest that skeletal muscle oxygenation levels in combination with augmented blood pressure response and central nervous system activation could be important determinants of exercise performance. Specifically, when the aforementioned parameters reach a predetermined maximal level, the exercise ability might be limited to maintain body’s integrity which in turn could modify the cardiovascular response or/and other physiological responses. This experimental approach could be used in the investigation of the mechanisms of cardiovascular dysfunction during exercise.
Reviewer’s comment
A sentence on line 145-148 and line 249-251 are duplicates.
Response:
Following the reviewer’s suggestion, we have deleted the duplicate in the revised manuscript (please see the revised manuscript in the materials and methods section (page 5, lines 160-161, and page 7, lines 256-257).
Reviewer’s comment
In general, the journal “Biology” is not a specific journal for research area of cardiorespiratory function.
Response:
We agree with the reviewer that the journal ‘Biology’ probably is not the most appropriate journal for cardiovascular responses to exercise. However, we would like to mention that this manuscript was submitted for the special issue of Biology entitled ‘Effects of physical exercise on human physiology and pathophysiology’ after the encouragement of the Editor.
Reviewer’s comment
Reviewer recommends putting a short sentence (interpretation) after each result section for better understanding.
Response:
Following the reviewer’s suggestion, we have provided a short interpretation sentence after each results section in the revised manuscript (please see the revised manuscript in the results section on pages 8-14, lines 311-496).
Reviewer’s comment
In fig1 A1-A4, please make sure the label of x-axis. Is this correct? Where is the muscle figure like brain fig 2 A1??
Response:
We agree with the reviewer that the muscle oxygenation figure during incremental cycle ergometer test without and with muscle blood flow restriction at isotime-isowatt (A) was not appropriate and we apologized for that. We have provided the correct muscle oxygenation figure (A) in the revised manuscript.
Reviewer’s comment
Please prepare the manuscript with more attention. For example, the abbreviation of “peak power output” (PPO) came up in lines 71, 87, 165. Also, in line 307, no definition of “TTE” before using. Furthermore, in this study, there are some key words like “relative comparison”, “absolute comparison”, “at rest”, “during exercise”, “at exhaustion” but in the text, it is hard to identify which one is what one. Also, sometimes authors used “increased” or “greater”, but it is not clear whether it is statistically different or not.
Response:
We agree with the reviewer’s concerns that the definition of the abbreviations should be consistent throughout the manuscript. Thus, we have defined all abbreviations before using and then we used only the abbreviation throughout the revised manuscript. Furthermore, as we have already mentioned in the materials and methods section (page 7, lines 296-300) the resting values referred to the average values for the time periods of the 6th to 9th minute during the 10-min resting period either without or with muscle blood flow restriction. During the incremental exercise mean values over the last 15 sec of each increment were used and at exhaustion was used the mean values over the last 15 sec of the exercise protocol. Finally, following the reviewer’s suggestion we have provided information about the significance of the changes throughout the revised manuscript.
Reviewer’s comment
In line 212, please put information about the company of “photoplethysmometer”.
Response:
Following the reviewer’s suggestion, we have provided this information in the revised manuscript. (revised manuscript, page 6, lines 224-225).
Reviewer’s comment
Please add comments regarding how does muscle oxygen level affect muscle contractile activity and induce muscle fatigue. Ultimately, facilitation of substrate depletion under low oxygen level ? or inadequate calcium uptake/release from SR/decreased pH or others?
Response:
We thank the reviewer for his/her comment. We would like to mention that muscle blood flow restriction has been shown to induce tissue hypoxia that could affect muscle contractile activity and as a result induce premature muscle fatigue. Especially, the accumulation of metabolic by-products such as lactate (La), phosphocreatine (PCr), inorganic phosphate (Pi), and hydrogen ion (H+), as an indication of inadequate O2 supply, had reported that increase during exercise with muscle blood flow restriction and consequently might inhibit the contractile process either directly or via metabolism, resulting in diminished exercise performance [Cairns 2006; Sports Med, 36, 279–291]. Indeed, acidosis may impair function of contractile properties by reducing: a) sarcoplasmic Ca2+ release and re-uptake, b) myofibrillar Ca2+ sensitivity, and c) activity of ATPase and key enzymes of glycolysis such as phosphofructokinase and phosphorylase [Kohmoto et al 1990; Circ. Res. 66, 622–632; Woodward, M et al 2018; Front. Physiol, 9, 862; Parolin, et al 1999; Am. J. Physiol. 277, E890–E900]. However, muscle fatigue is not caused by a single factor and various mechanisms are involved. Furthermore, in our study, the magnitude of force production decrease was similar without and with muscle blood flow restriction suggesting that muscle fatigue was identical between exercise protocols. When investigating skeletal muscle fatigue, muscle activation, vascular function, bioenergetics, changes in intracellular signaling and molecular mechanisms should all be considered, future studies should explore the effects of muscle blood flow restriction during exercise in all the above mechanisms.
Following the reviewer’s suggestion, we have provided the above information in the revised manuscript which now reads as follows (page 16, lines 566-576) ‘It is well accepted that muscle blood flow restriction during dynamic exercise induces intramuscular metabolic perturbation. Indeed, muscle blood flow restriction has been reported to augment the build-up of metabolic by-products such as lactate (La), phosphocreatine (PCr), inorganic phosphate (Pi), and hydrogen ion (H+) compared to control condition [74,75,76]. Furthermore, muscle blood flow reduction, mismatch of O2 delivery to metabolic demands, venous distention, and accumulation of metabolic by-products activate group III/IV muscle afferents [77,78,79]. Muscle afferents group III and IV activation appears to be essential for normal exercise hemodynamic and ventilatory responses and at the same time contribute to the development of peripheral fatigue and facilitate central nervous system (CNS) fatigue due to modulation of motor cortical output and inhibition of motoneuronal output during high-intensity exercise [25,38,39,80,81,82].’
Reviewer: 2
Reviewer’s comment
Comments and Suggestions for Authors
According to the simple summary, this study wants to find a new method to simulate hypoxic conditions as it might be caused by cardiovascular diseases.
Response:
As we mentioned before, this study wanted to explore the limiting factors of maximal oxygen uptake by comparing normal conditions with limited muscular oxygenation by applying cuff pressure on the exercising thigh. That was not clear in the initial manuscript due to our misunderstanding of what we stated in the simple summary. The purpose of the study is now clear throughout the revised version. We apologized for this misleading and have provided the above information in the revised manuscript in the simple summary (page 1, lines 13-26) which now reads as following: ‘This study aimed to elucidate whether exercise intolerance with muscle blood flow restriction was associated with increased central nervous system activation or/and deterioration in cardiovascular response. Muscle blood flow restriction (venous occlusion) during whole-body dynamic exercise accelerates a) the rate of skeletal muscle deoxygenation, b) the increase in systolic blood pressure, rating of perceived exertion, and cerebral activation without these variables to differ at exhaustion despite marked decrease in exercise time and maximal aerobic output was recorded. Maximal cardiovascular responses (i.e., heart rate, stroke volume, and cardiac output) were also significantly lower with muscle blood flow restriction. These findings suggest that skeletal muscle oxygenation levels in combination with augmented blood pressure response and central nervous system activation could be important determinants of exercise performance. Specifically, when the aforementioned parameters reach a predetermined maximal level, the exercise ability might be limited to maintain body’s integrity which in turn could modify the cardiovascular response or/and other physiological responses. This experimental approach could be used in the investigation of the mechanisms of cardiovascular dysfunction during exercise.
Reviewer’s comment
A plenty of typical cardiovascular parameters were randomly measured in order to find associations with blood flow restriction (BFR).
Response:
We agree with the reviewer that we have measured plenty of cardiovascular parameters, however, that was done for certain reasons well documented in the introduction and the methodology of the paper. Specifically, the purpose of the present study was to investigate whether the impairment of exercise performance with muscle blood flow restriction was associated with increased central nervous system activation or/and deterioration in cardiovascular response. In addition, thigh cuffs application is well known that induce venous occlusion, blood pooling and may restrict venous return to the heart which may contribute to exercise intolerance. Therefore, it is necessary to determine the whole cardiovascular response during exercise. Furthermore, muscle blood flow restriction during exercise has been demonstrated to exaggerate blood pressure response associated with peripheral vasoconstriction due to activation of group III/IV muscle afferents, a factor that may explain the impairment of exercise tolerance with muscle blood flow restriction. To confirm the above assumption, we measured the blood pressure response i.e., systolic, diastolic, and mean arterial blood pressure. Indeed, in the current study, we found that the blood pressure response was significantly higher during exercise with muscle blood flow restriction.
Reviewer’s comment
However, no reference to the aim of the study can be found as it was postulated in the simple summary in the entire residual manuscript, so it fails to answer the questions or to tell how and why this “new” method (its actually not new) can simulate CVD-conditions.
Response:
We accept the reviewer’s criticism, as we mentioned before the simple summary was not representative of the main purpose of this study and thus has not been mentioned anywhere in the manuscript (please see in the revised manuscript, page 1, lines 13-26). Furthermore, we would like to mention that this study is part of a larger project to propose a reliable method to focus exclusively on the role of starving for oxygen muscle without introducing whole-body hypoxemia. We have used the same experimental protocol at rest providing interesting and original data regarding that even at rest, the reduction of muscle oxygenation level could be sensed by the central system and alter cerebral activation (Appl Physiol Nutr Metab. 2021 Oct;46(10):1216-1224. doi: 10.1139/apnm-2020-1082). In the present study, we intend to make known the exercise data. It seems that muscle blood flow restriction would adversely affect the regulation of arterial blood pressure and cerebral oxygenation through an interactive talk between muscle and brain. Finally, this method induces, at rest and during exercise, in healthy subjects, muscular hemodynamic challenges similar to the ones observed in diseases such as hypertension and heart failure. We feel that this experimental paradigm could be extensively used in the future for studying the development of cardiovascular diseases. However, future studies must be conducted in order to establish it.
Reviewer’s comment
The study seems to be conducted appropriately and the paper is written in good language. Therefore, I can imagine that this research report might be interesting for readers (with moderate relevance).
Response:
We thank the reviewer for his/her positive comments on our work.
Reviewer’s comment
Substantial revisions have to be made in order to get the manuscript closer to a status which might be considerable for publication in my opinion.
Response:
We agree with the reviewer that the manuscript needs a throughout revision. Following the reviewers’ suggestions, we have provided a thorough revision of the manuscript (please see the revised manuscript).
Reviewer’s comment
Questions/comments:
Do the authors provide evidence how BFR simulates hypertension or heart failure?
Response:
We thank the reviewer for their comment. We would like to mention that the purpose of the present study was to investigate the effect of muscle blood flow restriction on hemodynamics, cerebral oxygenation, cerebral activation, and exercise performance during incremental exercise to exertion. Specifically, we elucidated whether the impairment of exercise performance with muscle blood flow restriction was associated with failure of central nervous system activation or/and deterioration in cardiovascular response. Therefore, it was outside the purpose of this study to provide evidence of how muscle blood flow restriction during exercise simulates hypertension and heart failure.
It is worth mentioning that the muscle blood flow restriction method appears to induce physiological responses similar to those observed in chronic diseases such as hypertension, heart failure, and peripheral vascular disease. Specifically, patients with cardiovascular diseases such as hypertension, heart failure, and peripheral vascular disease characterized by exaggerated muscle afferents group III and IV feedback with substantial sympathoexcitation and blood pooling in capacitance vessels that compromise ventricular contractility, muscle blood flow and O2 delivery and cerebral blood flow as well during exercise, resulting in exertional dyspnoea and exercise intolerance (Sullivan et al. 1989; Circulation. 80(4): 769-81; Amann et al. 2014; Int J Cardiol.174(2): 368-75; Piepoli and Crisafulli 2014; Exp Physiol. 99(4): 609-615. Smith et al. 2020; Exp Physiol.105(5): 809-818; Smith et al 2020; J Physiol. 598(23): 5379-5390; Rehan 2021; J Physiol. 599(3): 733-734; Anguis & Crisafulli 2020; Eur J Prev Cardiol. 27(17):1862-1872). Similarly, muscle blood restriction during exercise, in the current study, reduces arterial blood flow to muscles and muscle oxygenation, restricts venous blood flow, causes blood pooling in capacitance vessels distal to the cuff, exaggerates blood pressure response associated with peripheral vasoconstriction due to activation of group III/IV muscle afferents (exercise pressor reflex), impairs cardiovascular response resulting in enhance cerebral activation and exercise intolerance. Thus, there is some evidence in support of using the muscle blood flow restriction intervention to investigate the development of cardiovascular diseases and needs to be further investigated.
Reviewer’s comment
Does Venous occlusion (this is actually what you reach when you occlude with 120mmHg really causes hypoxic conditions?
Response:
We thank the reviewer for his/her comment. In the current study, we applied a cuff occlusion pressure similar to the systemic perfusion pressure to affect venous return, decrease arterial blood inflow without compressing the arteries, and reduce muscle oxygenation. The occlusion pressure of 120 mmHg using the wide cuffs (18 cm width) has been suggested to induce venous occlusion i.e., restrict venous blood flow and cause pooling of blood in capacitance vessels distal to the belt while restricting arterial blood flow as well (Mouser et al 2017, Eur J Appl Physiol, 117(7), 1493-1499).
Hypoxic condition is defined as a condition where tissue oxygen tension or/and mitochondrial oxygen tension is below normal due to failure of oxygenation at the tissue level as a consequence of oxygen delivery to tissue impairments or inability of the tissues to use oxygen effectively (Samuel & Franklin 2008; doi.org/10.1007/978-0-387-75246-4_97). Unfortunately, in the present study, it was not feasible to measure tissue oxygen tension during exercise with muscle blood flow restriction. However, several studies that applied venous occlusion (cuff occlusion pressure 60-220 mmHg) during resistance and aerobic exercise protocols have reported augmented metabolic response to blood flow restriction exercise as confirmed by the accumulation of metabolic by-products, such as lactate, di-protonated phosphate, deoxygenated hemoglobin, inorganic phosphate, and hydrogen ions that usually accompanies tissue hypoxia (Freitas et al 2021; Front. Physiol. 12: 747759; Yasuda et al 2010; Metabolism. 59(10): 1510-1519; Takarada et al 2000; J Appl Physiol. 88: 61-65). In the current study, we estimated that an occlusion pressure of 120 mmHg could reduce arterial blood flow approximately by 55-65% (Hunt et al 2017; Eur J Appl Physiol. 116(7): 1421-1432) and as a consequence could impair O2 delivery to tissue and induce tissue hypoxia. Furthermore, we found that the concentration of deoxy-hemoglobin and (Δ[HHb]) and hemoglobin difference (Δ[HbDiff]) was significantly affected during exercise with muscle blood flow restriction. Collectively, we support that the occlusion pressure of 120 mmHg could induce tissue hypoxia.
Reviewer’s comment
There is a plenty of Literature which describes local muscle oxygenation during venous occlusion. So it is not surprising that it happens the same in your study. Why don’t you cite any of those studies?
Response:
We would like to mention that only a few studies had investigated the effects of either venous or arterial occlusion on skeletal muscle oxygenation [Takarada et al 2000; J Appl Physiol, 88(6), 2097-2106; Geladas et al 2010; The Open Sports Med J 2010, 4, 9-16; Ganesan et al 2015; Med Sci Sports Exerc, 47(1): 185–93; Pearson, et al 2015, Sports Med, 45(2), 187–200; Valenzuela et al. 2019, Int J Sports Physiol Perform, 1280–1287; Willis et al 2018; Physiol Rep, 6(19): e13872; Willis et al 2019; Eur J Appl Physiol. 119(8): 1819–28; Peyrard et al 2019; Eur J Appl Physiol, 119(7): 1533–1545; Willis et al 2019; J Sci Med Sport, 22(10): 1151-1156; Wei et al 2021; Biol Sport 2021, 38(3), 437-443]. However, the majority of such studies have been conducted during resistance exercises or/and repeated sprint trials to investigate the possible mechanisms involved in skeletal muscle adaptations when muscle blood flow restriction is combined with resistance or repeated sprint exercise protocols. This exercise mode was different from cycling incremental exercise to exertion used in the current study. Thus, the effect of muscle blood flow restriction on muscle oxygenation during whole-body dynamic exercise is not clear and for this reason, we did not refer to the majority of these studies. In the revised study we have added only the following references that are more relevant to the purpose of our study:
- Takarada, Y.; Takazawa, H.; Sato, Y.; Takebayashi, S.; Tanaka, Y. Effects of resistance exercise combined with moderate vascular occlusion on muscular function in humans. J Appl Physiol 2000, 88(6), 2097-2106. doi: 10.1152/jappl.2000.88.6.2097.
- Wei J.; Nassis GP.; Gu, Z.; Zou, Y.; Wang, X.; Yongming Li, Y. Acute physiological and perceptual responses to moderate intensity cycling with different levels of blood flow restriction. Biol Sport 2021, 38(3), 437-443. doi: 10.5114/biolsport.2021.100146.
- Pearson, S.J.; Hussain, S.R. A review on the mechanisms of blood-flow restriction resistance training-induced muscle hypertrophy. Sports Med 2015, 45(2), 187–200. doi: 10.1007/s40279-014-0264-9.
Reviewer’s comment
A lot of cited articles but a lot of them are pretty basic and overlapping.
Response:
We agree with the reviewer that we have mentioned several references that are related to the limiting factors of exercise capacity and some of them are overlapping. In the revised manuscript we have deleted many of them.
Reviewer’s comment
l.314, l.355: Please avoid the use of the wording from your statistic software (e.g. “time” - you did not measure the influence of time but the influence of intensity). Please describe what the factor means instead.
Response:
Following the reviewer’s suggestion, we have removed the term ‘time’ from the revised manuscript. (revised manuscript in results section, pages 8-14, lines 311-496). Furthermore, we have described the factors used in ANOVA analysis in the revised manuscript in the data analysis and statistics section (page 7, lines 298-305).
Reviewer’s comment
There are way too many parameters which do not contribute to answer the research question. Please remove all unnecessary variables to increase readability. For example, PetO2 and PetCO2 do not matter here, but also many more. Further EEG and EMG seem to be off topic here. It would only make sense here if you would have provided significant evidence that there might be an connection between BFR and those parameters. The plenty of irrelevant variables is just distracting.
Response:
We agree with the reviewer’s opinion that the lower the number of variables presented the better the readability of the manuscript. However, the lower the number of variables presented the more difficult becomes to explain the phenomenon. For example, PetO2 and PetCO2 fluctuations during exercise are used as an indication of hypoventilation or hyperventilation which may affect the exercise tolerance per se. Moreover, it is well known that cerebral blood flow or/and cerebral oxygenation are highly sensitive to oxygen and carbon dioxide fluctuation [Rasmussen et al 2006; Eur J Appl Physiol. 96(3): 299-304; Zhang et al 2011; J Appl Physiol. 110(2): 352-358; Ainslie et al 2008; Am J Physiol Regul Integr Comp Physiol 295(5), R1613-1622; Iwasaki et al 2011; J Cereb Blood Flow Metab. 31(1): 283-292]. Therefore, cerebral blood flow and oxygenation measurements during exercise should be accompanied by measurements of the PetO2 and PetCO2 in order to provide evidence to explain the variation of cerebral blood flow and cerebral oxygenation.
Furthermore, we would like to mention that the purpose of this study was to investigate whether exercise intolerance with muscle blood flow restriction was associated with increased central nervous system activation or/and deterioration in cardiovascular response. It is well accepted that electroencephalography (EEG) assesses the cerebral electrocortical activity in specific brain areas providing essential information about the central nervous system activity [Kumar & Bhuvaneswari 2012, Procedia Engineering 38: 2525-2536; Nielson & Nybo 2003; Sports Med 33(1): 1-11; Nielsen et al 2001; Flugers Arch. 442(1): 41-48]. In the current study, we found that cerebral electrocortical activity was identical at task failure without and with muscle blood flow restriction despite marked differences in time to exhaustion and peak power output. These data suggest that the rate of cerebral activation increase was exaggerated with muscle blood flow restriction, and this could be a significant determinant of exercise tolerance. In addition, muscle activation measured via EMG amplitude provides an indirect indication of motoneuronal output to the working muscle and an indirect index of fatigue development during exercise revealing changes in the rate of motor unit recruitment and firing rate [Finsterer & Mahjoub 2014; Am J Hosp. Palliat Care. 31(5): 562-75; Del Vecchio et al 2017, J Appl Physiol123(4): 835-843; Riley et al 2008, Muscle Nerve. 37(6): 745-53; Potvin J 1991; J Appl Physiol. 82(1): 144-151; 4. Suzuki et al 2002; Med Sci Sports Exerc. 34(9): 1509]. In the current study, iEMG activity was progressively increased during exercise without being different between experimental conditions at task failure despite marked differences in exercise time and peak power output. Thus, the central motor drive was identical and maximal without and with muscle blood flow restriction at exhaustion. Accordingly, we have provided a throughout revision of the manuscript providing considerable evidence of these variables with the muscle blood flow restriction and the possible effect on exercise intolerance.
Reviewer’s comment
Fig 2 - Where are the errorbars in the A-panels in the end? I have major concerns regarding the cerebral NIRS data here. Can you explain the severe drop/increases here?
Response:
Reviewer’s point sounds valid but the absence of the errorbars at the last absolute workload during exercise without and with muscle blood flow restriction needs further clarification. We would like to mention that the participants were recreational cyclists or triathletes and their exercise capacity during incremental exercise trials without and with muscle blood flow restriction was different among participants. Therefore, only one participant was able to achieve the 390 watts during exercise without muscle blood flow restriction and the 300 watts during exercise with muscle blood flow restriction and as a result was not able to calculate the standard error of measurements. In order to take into consideration, the reviewer’s concern we have deleted the data of the last workload in Figure 2 and we have mentioned the number of participants that reach each workload.
The large increase or decrease in cerebral oxygenation indices observed at the last workload is probably due to the variability observed in this one individual. We followed the same procedure for Figure 1, which demonstrates a similar pattern.
Reviewer’s comment
The first sentence of the discussion does not refer to the research question in the short summary
Response:
As we mentioned before the simple, not short, summary was not representative of the main purpose of this study. The purpose of this study was to investigate the effect of reduced skeletal muscle oxygenation on hemodynamic responses, cerebral oxygenation, and cerebral activation during incremental exercise to exertion in healthy individuals. Furthermore, we elucidated whether the impairment of exercise performance with muscle blood flow restriction was associated with failure of central nervous system activation or/and deterioration in cardiovascular response. We have provided the above information in the simple summary section in the revised manuscript (page 1, lines 13-26). Furthermore, we have revised the discussion section to avoid misunderstandings and confusion.
Reviewer’s comment
4.1 couldn’t it be that VO2max is reduced because the lowered PPO required less energy and that exercise was rather terminated because of discomfort?
Response:
We agree with the reviewer that when the power output is reduced, sooner or later the oxygen consumption will be affected. It is well known that there is a close linear relationship between VO2 and workload. Indeed, in our study, we found a significant correlation between VO2max and PPO in NC (r=0.639, p=0.019) and WC (r=0.587, p=0.035) conditions. However, the reduced peak power output seems to be attributable to muscle oxygen delivery impairment due to venous occlusion and blood pooling in the lower extremities, which enhanced cerebral activation and perceptual sensation of effort.
Reviewer’s comment
Also, HRmax WC is most likely reduced because of less energy requirement.
Response:
We thank the reviewer for his/her comment. We would like to mention that, in our study, the reduction of maximal heart rate seems to contribute to the lowering of maximal cardiac output during exercise with muscle blood flow restriction and then exercise intolerance. As we mentioned, one hypothesis that we stated was that the reduced HRmax may simply be the result of a lower maximal workload and VO2max. This hypothesis states that HR is detected by VO2 or/and workload at any level of exercise. Certainly, there is a tight linear relationship between HR and VO2 or/and workload from rest to maximal exercise. However, several studies that manipulated the heart rate response to incremental exercise to exhaustion through acute hypoxia and hyperoxia, as well as chronic systemic hypoxia, sympathetic or/and parasympathetic blockade reported that maximal heart rate does not always follow the VO2 and workload fluctuations (Bogaard et al 2002; Boushel et al 2001; Hopkins et al 2003; Amann et al 2006a,b). Furthermore, the underlying mechanisms that HR may tie to VO2 or/and workload are not clear. In our study, we observed that the restricted skeletal muscle oxygenation did not affect HR at rest, increased HR during exercise at the same absolute workload, and significantly reduced HR at exhaustion. At the same time cerebral activation, sympathetic nervous system activation, skeletal muscle deoxygenation and oxygenation index, perceptual response, peripheral muscle fatigue, and systolic blood pressure were identical between the two exercise protocols at task failure despite marked differences in time to exhaustion, peak power output, and VO2max. In addition, we did not find any significant correlation between HRmax and peak power output either in NC (r=0.153, p=0.617) or WC (r=0.449, p=0.124) conditions. It seems that the reduced HRmax was not caused simply by the lower power output produced.
We have provided this information in the revised manuscript (page 18, lines 685-698) which now reads as following: ‘It appears that the reduced HRmax with thigh cuffs application seems to contribute to the reduction in max and the subsequent exercise intolerance. However, the underlying mechanisms are still unknown. Reduced HRmax may simply be the result of a lower maximum workload. It is well known that there is a close linear relationship between HR and workload from rest to maximal exercise. However, several studies that manipulated the heart rate response to incremental exercise to exhaustion through acute hypoxia and hyperoxia, as well as chronic systemic hypoxia, sympathetic or/and parasympathetic blockade reported that maximal heart rate does not always follow the O2 and workload fluctuations [12,90,91,92]. In our study, we observed that restricted skeletal muscle oxygenation had no effect on HR at rest, increased HR during exercise at the same absolute workload, and significantly reduced HR at exhaustion. In addition, we found no significant correlation between HRmax and PPO in NC (r=0.153, p=0.617) and WC (r=0.449, p=0.124) conditions. It seems that the reduced HRmax was not caused simply by the lower power output produced.’
Reviewer’s comment
General points:
Focus is missing in this manuscript. You did not introduce a new method for simulating CVD-induced hypoxic conditions. The discussion needs to be completely rewritten (without irrelevant parameters). The saved space should be invested to find mechanistic explanations for the observed effects. A story line should be persuaded throughout the manuscript.
Response:
We thank the reviewer for his/her comment. Following the reviewer’s suggestions, we have revised the entire manuscript that local occlusion model has some advantages to reveal mechanisms of physiological events.